# Intensity–Duration–Frequency curves from remote sensing rainfall estimates: comparing satellite and weather radar over the Eastern Mediterranean

Francesco Marra[1], Efrat Morin[1], Nadav Peleg[2], Yiwen Mei[3], Emmanouil N. Anagnostou[3]

[1]Institute of Earth Sciences, Hebrew University of Jerusalem, 91904, Israel
[2]Institute of Environmental Engineering, Hydrology and Water Resources Management, ETH Zurich, Switzerland
[3]Department of Civil and Environmental Engineering, University of Connecticut, Storrs, CT, USA

*Correspondence to*: Francesco Marra (marra.francesco@mail.huji.ac.il)

**Abstract.** Intensity–Duration–Frequency (IDF) curves are widely used to quantify the probability of occurrence of rainfall extremes. The usual rain gauge based approach provides accurate curves for a specific location, but uncertainties arise when ungauged regions are examined or catchment scale information is required. Remotely sensed rainfall records, e.g. from weather radars and satellites, are recently becoming available, providing high resolution estimates at regional or even global scales: their uncertainty and implications on water resources applications urge to be investigated. This study compares IDF curves from radar and satellite (CMORPH) estimates over the Eastern Mediterranean (covering Mediterranean, semiarid and arid climates) and quantifies the uncertainty related to their limited record on varying climates. We show that radar identifies thicker tail distributions than satellite, in particular for short durations, and that the tail of the distributions depends on the spatial and temporal aggregation scales. The spatial correlation between radar-IDFs and satellite-IDFs is as high as 0.7 for 2–5 years return period and decreases with longer return periods, especially for short durations. The uncertainty related to the use of short records is important when the record length is comparable to the return period (~50%, ~100% and ~150% for Mediterranean, semiarid and arid climates, respectively). The agreement between IDF curves derived from different sensors on Mediterranean and, to a good extent, semiarid climates, demonstrates the potential of remote sensing datasets and instils confidence on their quantitative use for ungauged areas of the Earth.

## 1 Introduction

Intensity–Duration–Frequency (IDF) curves are widely used in hydrological design and as decision support information in flood risk and water management (Watt and Marsalek, 2013). They are distribution functions of rain intensity maxima conditioned on duration and allow linking the characteristics of a rainfall event to the probability of its occurrence (Chow et al., 1988; Eagleson, 1970). Derivation of IDF curves generally relies on historical precipitation data and consists in fitting an extreme value distribution to extreme rainfall values. Rain gauges, providing long records of precipitation data, are traditionally used to estimate IDF curves at the gauge locations. Nevertheless, owing to the sparseness of gauge networks worldwide (Kidd

et al., 2016), this approach raises important issues when design applications require information at the catchment scale. Moreover, the representativeness of gauge derived IDFs decreases moving away from the gauge location and no or very sparse information is available for the many ungauged locations of the Earth.

In the last decades a body of research has been devoted to these issues. Areal reduction factors and design storms assume homogeneity of rainfall extremes climatology to adapt the point IDF values estimated by rain gauges to wider areas, such as catchments, basing either on the climatology of the region (Sivapalan and Bloschl, 1998), on spatial precipitation observations (Bacchi and Ranzi, 1996; Durrans, 2002; Allen and De Gaetano, 2005; Lombardo 2006; Overeem et al., 2010; Wright et al., 2014) or stochastic model simulations (Peleg et al., 2016, 2017). In principle, areal reduction factors may depend on a number of factors, such as geographic location, characteristics of the examined catchment, analysed duration and period, season, meteorological conditions, and others (Svensson and Jones, 2010). Their derivation is thus hampered by many sources of uncertainty. Regionalization and interpolation techniques continue assuming spatial homogeneity to derive IDF curves for ungauged locations (Ceresetti et al., 2012). However, homogeneity of rainfall extremes is a weak assumption when wide areas are considered or when, even at the small scales, the data record length is limited, especially in the case of short durations (Peleg et al., 2016).

Remote sensing instruments, such as weather radars or satellites, provide high spatiotemporal resolution (i.e. 1–10 km and 5–60 min), distributed, regional or even global rainfall estimates. These datasets allow capturing dynamics and variability of rainfall extremes at scales that cannot be represented by rain gauges (Amitai et al., 2011; Chen et al., 2013; Marra et al., 2016; Panziera et al., 2016; Tapiador et al., 2012) and permit to overcome the issues related to the conversion of point to areal information. Given the quantitative uncertainty related to remote sensing precipitation products (Berne and Krajewski, 2013; Mei et al. 2014; Stampoulis et al., 2013) and their limited records, unable to provide adequate samples of the climatology, the possibility of using such datasets for rainfall frequency analysis has only recently started to be explored. Main focus of these studies was the assessment of their potential (Eldardiry et al., 2015), for poorly instrumented regions (Awadallah et al., 2011; Nastos et al., 2013) and areas characterized by strong climatic gradients (Marra and Morin, 2015). Few authors, so far, provided quantitative IDF curves from remote sensing instruments, using some kind of regionalization approach (Endreny and Imbeah, 2009; Overeem et al., 2009; Wright et al., 2013; Panziera et al., 2016).

Since quantitatively accurate information is essential for design applications and the derivation of IDF curves based on historical records does not require short latency in the data, these studies made use of gauge-adjusted products and assessed the accuracy of the IDF curves derived from remote sensing datasets using rain gauge curves as a reference. However, this approach neglected two important aspects. First, early warning systems, e.g. for flash floods (Borga et al., 2011; Villarini et al., 2010; Borga et al., 2014), urban floods (Yang et al., 2016), landslides/debris flows (Tiranti et al. 2014; Borga et al., 2014; Segoni et al., 2015) or heavy rain (Panziera et al. 2016), need to operate in real-time and rely on short-latency remote sensed measurements. In these situations, calculating the frequency of near real-time estimates using IDF curves derived from gauge-adjusted data could provide misleading results. It is therefore useful to analyse the characteristics of IDF curves derived from non-adjusted rainfall data, which are expected to represent the frequencies of near real time estimates. Second, areal IDFs

provided by remote sensing instruments are expected to differ from point IDFs (Peleg et al., 2016), and the use of different records (i.e. different samples of the climate) introduces further differences. No exact match between remote sensing and rain gauge IDFs should be expected a priori. Moreover, evaluating unadjusted data is important for demonstrating the value of satellite products in ungauged areas.

The aim of this study is to advance knowledge on the use of remote sensing precipitation estimates for rainfall frequency analysis. IDF curves computed from different datasets, namely the National Oceanic and Atmospheric Administration (NOAA) Climate Prediction Center morphing (CMORPH) technique (Joyce et al. 2004), the gauge–adjusted CMORPH (Xie, et al., 2011) and the ground based C-Band Shacham weather radar archive (Marra and Morin, 2015), are compared over the Eastern Mediterranean using common spatiotemporal scales and records. The effect of spatial and temporal aggregation of rainfall is analysed. The uncertainty related to the use of short records in the climatic contexts characterizing the study region is assessed using long records of rain gauge observations. To the authors' knowledge this is the first study in which at-site IDF curves derived from different gridded remote sensing datasets are compared.

The study area and the rainfall datasets are presented in section 2. Section 3 describes the methods used for computing and comparing satellite and radar IDF curves and for estimating the uncertainty related to the record length. In Sect. 4 the results of the study are presented and discussed. Section 5 provides the conclusions and suggestions for the future use of remote sensing datasets for rainfall frequency analysis.

## 2 Study area and data

### 2.1 Study area

The present study is focused on the Eastern Mediterranean, in particular on the area covered by the Shacham weather radar, shown in Fig. 1. The orography presents a longitudinal organization that, going east from the Mediterranean Sea, encounters a coastal plain, a hilly region (up to ~800 m a.s.l.), the Jordan rift valley (~ -400 m a.s.l.) and the Jordan Plateau (~1000 m a.s.l.). Mountains in the north of the study area raise up to 2800 m a.s.l.

Three climatic regions, Mediterranean, semiarid and arid, can be identified in the area, corresponding to the Csa, BSh and BWh Koppen-Geiger definitions, respectively (Peel et al., 2007). The criteria used to define these classes are reported in Table 1. In this study, we follow the classification by Srebro and Stoffer (2011). Mediterranean climate characterizes the north-western coastal region and the northern part of the Jordan Plateau. Arid climate characterizes the south-western and southern portions of the area, and the Jordan rift valley. These two regions are separated by a strip characterized by semiarid climate that can sometimes be very narrow (Fig. 1). Climatic gradients are important with mean annual precipitation exceeding 1000 mm yr$^{-1}$ in the northern part of the area and dramatically dropping from 600 to 100 mm yr$^{-1}$ in just 25 km distance along the narrow strip west of the hilly region (Alpert and Shafir, 1989; Goldreich, 1994). The summer is dry, with almost no rain from June to August. Rainfall over the Mediterranean areas is generally brought by cold fronts and postfrontal systems, associated with mid latitude cyclones (Goldreich, 2003; Peleg and Morin, 2012), and by Syrian low and active Red Sea Trough,

occasionally correlated with flash floods, in the semiarid and arid climates (Dayan et al., 2001; Dayan and Morin, 2006; Kahana et al. 2002; de Vries et al., 2013). On rare occasions the Subtropical Jet, or Tropical Plume, brings widespread rainfall over the whole region (Dayan and Morin, 2006; Kahana et al., 2002; Rubin et al., 2007; Tubi and Dayan, 2014; Dayan et al., 2015). Important gradients have been reported also for the climatology of extreme rainfall. Low return period intensities were found
to be scaled with the mean annual precipitation. Conversely, the more arid the climate is, the more skewed the extreme value distribution is, with long return period intensities for arid areas being higher than the corresponding values for semiarid and Mediterranean areas, especially for short durations (Ben Zvi, 2009; Marra and Morin, 2015).

## 2.2 Radar data

The Shacham weather radar is a C-Band (5.35 cm wavelength) non-Doppler instrument, operational since the late 1980s.
Observations from this radar have been archived from October 1990 to Mar 2014, providing a unique 23-year record and have been extensively used for climatologic and hydrological studies (e.g. Morin et al. 1995, 2001, 2009; Morin and Gabella 2007; Peleg and Morin 2012; Peleg et al. 2013, 2016; Yakir and Morin 2011).

The Shacham radar record has recently been reanalysed for rainfall frequency analysis (Marra and Morin 2015). Radar quantitative precipitation estimation is obtained combining physically based correction algorithms and quantitative
adjustments based on the comparison with rain gauge measurements. The procedure is discussed in details in Marra and Morin (2015) and included: (i) checking of antenna pointing, (ii) ground clutter filtering, corrections for the effects of (iii) wet radome attenuation, (iv) attenuation due to the propagation of the radar beam, (v) beam blockage and (vi) vertical variations of reflectivity, together with a (vii) hail filter. A two-step bias adjustment was then applied combining a yearly range dependent and an event based mean field bias adjustments based on comparison with quality checked rain gauge data. Radar rain rates
exceeding 150 mm h$^{-1}$ have been set to this cap value in order to avoid contaminations due to hail. The readers are referred to Marra et al. (2014) and Marra and Morin (2015) for an extensive description of the correction procedures and the quantitative assessment of the archive.

This study is based on the hourly radar archive. Each hourly radar product is created when at least 60% of radar scans are available during the 1 h time interval. This quality check on the data availability ensures a good coverage of the examined time
interval and allows exploring durations longer than the ones analysed by Marra and Morin (2015).

## 2.3 Satellite data

The satellite precipitation products selected for this study are the high resolution CMORPH (HRC) and its gauge-adjusted version (CHRC) available from NOAA CPC. The two products, with a resolution of 0.073°/half-hourly, offer high resolution, quasi global, long term records of satellite rainfall estimates (available, as at November 2015, for 16 years: from January 1998
to December 2013). CMORPH integrates multiple satellite based microwave rain estimates (Ferraro et al., 1997; 2000; Kummerow et al., 2001) in space and time using motion vectors derived from infrared images (Joyce et al. 2004). The newly available gauge-adjusted CMORPH applies daily gauge-adjustment using estimates from ~30,000 gauges worldwide (Xie, et

al., 2011). The gauge-adjusted CMORPH product is using gauge data from ~12 gauges in the region (Chen et al., 2008), which may be also be used in the adjustment of the radar-rainfall dataset. The half-hourly original data has been aggregated to hourly intervals to match the radar archive resolution.

## 2.4 Rain gauge data

The small scale representativeness of rain gauge measurements makes them not suitable for a large scale quantitative assessment of remote sensing products (Gires at al., 2014; Peleg et al., 2016). Here, we take advantage of their long records to empirically quantify the uncertainty in rainfall frequency analysis related to the use of short records in different climatic conditions.

We identified 11 rain gauges, operated by the Israeli Meteorological Service, among the ones already used by Marra and Morin
(2015). Rain gauge data is available as storm maxima values extracted from digitized charts for durations up to four hours (before 2000) or automatic measurements with 10 min (from 2000 up to 2005) and 1 min (from 2006 on) resolution. This limits our analysis of rain gauge data to durations shorter than four hours. When possible, rain gauge data have been aggregated into hourly blocks to be as consistent as possible with the remote sensing datasets. Stationarity of the records has been tested at 0.1 significance level (Phillips and Perron, 1988). The selected rain gauges belong to different climatic regions with five,
four and two rain gauges available for Mediterranean, semiarid and arid climates, respectively (Fig. 1). The length of the analysed records ranges between 30 and 67 years and is at least double of the satellite datasets record (16 years) with just one exception (30 years) belonging to the semiarid climatic region.

## 3 Methods

### 3.1 Derivation of Intensity–Duration–Frequency curves

In this study, the Generalized Extreme Value (GEV) distribution (Appendix A) is used to fit the Annual Maxima Series (AMS) of average rain intensities observed over 1, 3, 6, 12 and 24 h durations. The use of AMS ensures independency of the elements of the series and, rather than peak over threshold series, is suitable for this study because it does not require the definition of thresholds, problematic operation on highly variable climatic conditions and potentially undermining the interpretation of the comparison between different datasets. The GEV distribution is a three parameters extreme values distribution used worldwide
to model rainfall extremes (Fowler and Kilsby, 2003; Gellens, 2002; Koutsoyiannis, 2004; Overeem et al., 2008). It is described by the location, scale and shape parameters, representing mean, dispersion and skewness of the distribution, respectively. The GEV distribution has been shown to fit the AMS better than other distributions, such as the Pearson type III or the generalized logistic (Alila, 1999; Kysely and Picek, 2007; Perica et al., 2013; Schaefer, 1990). It's being widely used for rainfall frequency analyses based on remote sensed data (Eldardiry et al., 2015; Marra and Morin, 2015; Overeem et al., 2009; Paixao et al.,
2015; Panziera et al. 2016; Peleg et al, 2016), owing to its ability to include all the three asymptotic extreme value types (Gumbel, Fréchet and Weibull) into one (Katz et al., 2002).

In order to have radar and satellite data on a common grid suitable for the comparison, the full archive of hourly radar data was remapped by spatially averaging the $1\times1$ km$^2$ radar pixels to the corresponding ~$8\times8$ km$^2$ CMORPH pixels. We identified the AMS of calendar years for the examined durations using a moving window with 1 h jumps. Even if using hydrologic years is more natural, we chose to use calendar years to exploit the full CMORPH record, available from January 1998 to December 2013. A minimum time lag of 48 h between annual maxima observed in different years was set to fulfil the independency requirements of the GEV theory. All the available rainfall estimates have been included in the IDF estimation, even if data from the other sources was missing during a given storm. No co-occurrence of the annual maxima is thus imposed. There are, in fact, potential situations in which radar or rain gauges missed storms due to technical problems or power issues that cannot be directly identified from the data itself (Morin et al., 1998; Ben Zvi, 2009; Marra and Morin, 2015). At-site GEV parameters (i.e. pixel by pixel) were identified using the maximum likelihood method (MATLAB statistics toolbox). Situations for which the fitting was not successful due to convergence problems or excessive number of iterations were discarded. Less than 0.7% of the cases were discarded for this reason. The IDF values were calculated for a set of five return periods (2, 5, 10, 15 and 25 years).

The effect of spatial and temporal aggregation of rainfall estimates, key issue when dealing with remote sensing instruments, is analysed spatially aggregating the original radar record (23 years, $1\times1$ km$^2$) on grid sizes gradually increasing from $2\times2$ to $64\times64$ km$^2$ and for durations from 1 to 24 h using moving windows with 1 km and 1 h jumps. In this analysis, radar is preferred over CMORPH due to its ability to provide more direct rainfall estimates at the small spatial and temporal scales.

## 3.2 Comparison between Intensity–Duration–Frequency maps

IDF maps obtained from the satellite precipitation datasets (HRC and CHRC) are compared to the ones obtained from the radar archive during corresponding years (16 years: 1998–2013). The comparison is extended over an analysis domain defined excluding the pixels that are known to be not reliable. In particular, pixels located closer than 27 km or farther than 185 km from the radar or behind the hilly region are excluded due to insufficient reliability of the radar data, and pixels located in proximity of major lakes are excluded due to false rainfall signals in the CMORPH estimates (e.g. Guo et al., 2015). The number of pixels analysed for each climatic region is reported in Table 1. A limited number of cases with problematic data, providing unrealistic outcomes, could still be found in all the products. In order to avoid single problematic situations dominating the results, we analysed the distribution of the GEV parameters over the whole study area and excluded from the comparison the pixels for which any of them was outside of the 1–99[th] quantiles interval of the corresponding distribution.

Three widely used non-dimensional, normalized metrics are selected to compare radar-IDF and satellite-IDF maps: Correlation Coefficient (CC), measuring the spatial correlation of the maps; multiplicative bias (Bias), measuring the mean quantitative agreement of the maps; and Normalized Standard Difference (NSD), measuring the variability of the residuals of the normalized maps. Additional information on the metrics is provided in Appendix B.

**3.3 Uncertainty related to the record length**

We assumed the records of rain gauge data to be a complete sample of the climatology of extremes for return periods comparable to the remotely sensed data record length. Synthetic records of rain gauge data were created by randomly sampling years, without replacement, from the full rain gauge record, and the corresponding IDF values were calculated. We focused on synthetic records of 10, 15, 20 and 25 years and bootstrapped the operation 999 times for each rain gauge and for each synthetic record length. The 5–95[th] quantile interval of the obtained distributions was used to measure of the uncertainty related to the record length (Overeem et al., 2008).

**4 Results and discussion**

**4.1 Comparison between satellite- and radar- derived GEV parameters**

The distribution of the GEV parameters (25–75[th] quantile intervals and median over the whole study area) derived from radar and satellite datasets in the three climatic regions and over the Mediterranean Sea (sea, from here on) are presented in Fig. 2. We recall here that the scale, location and shape parameters provide a measure of the mean, dispersion and skewness of the underlying distribution, respectively. Location parameters from HRC (CHRC) estimates are smaller (larger) than the ones from radar over Mediterranean climate and over the sea, meaning that extreme values from HRC (CHRC) are in general lower (higher) than radar extreme values while in semiarid and arid climates HRC and CHRC generally identify higher parameters than the radar (i.e. higher extreme values). Differences in the location parameters can be associated to the bias between extreme values in the datasets. The scale parameters are normalized over the corresponding location parameters to appreciate the relative differences. Normalized scale parameters from HRC and CHRC are similar and lower than the ones derived from radar. Normalized scale parameters, together with their variability, tend to increase when moving from sea to Mediterranean, semiarid and arid climates. The drier climate, the larger the dispersion of the GEV distribution. A slight increase of the normalized scale parameters with duration can be noticed in the HRC/CHRC data.

The shape parameters are mostly greater than zero, suggesting, in line with previous studies (Katz et al., 2002; Papalexiou and Koutsoyiannis, 2013), that type II extreme values distribution should be considered for the area. Both radar and satellite products derive high shape parameters for arid climate that decrease when moving to semiarid and to Mediterranean climates, thus suggesting that drier regions are characterized by thicker tailed distributions. Over the sea, satellite shape parameters are close to zero, while radar shape parameters tend to be higher. Increasing duration, radar-derived parameters are decreasing, while satellite-derived parameters tend to increase, becoming comparable for durations longer than ~6 h. In general, the shape parameters derived from radar are higher than the ones derived from CMORPH so that, in arid climate, the radar dataset includes few cases with a shape parameter greater than 0.5 and, in single occasions, even greater than 1. Among the many possible causes, this can be associated to the measurement uncertainty of radar estimates and to the record intermittency due to radar shutdowns (Marra and Morin, 2015). In fact, the rainfall regime of arid climate is characterized by short and intense

events (Karklinsky and Morin, 2006), so that gaps in the record can cause the missing of important storms leading to particularly low estimates of some annual maxima and, consequently, to thicker-tailed distributions. It should also be noted that missing of short duration extremes by CMORPH due to the overpasses frequency of microwave satellites could contribute to the differences observed between CMORPH and radar.

### 4.1.1 Effect of spatial and temporal aggregation

The location and scale parameters consistently decreased as the spatial and temporal aggregation scales increased. This is an expected effect, caused by the smoothing of rainfall fields operated by the spatial averaging, therefore results are not reported in this paper. Conversely, it is interesting to analyse the shape parameter. The distributions of the shape parameters derived from the full radar record aggregated on grid sizes increasing from $2\times2$ to $64\times64$ km$^2$ and on durations increasing from 1 to 24 h are presented in Fig. 3. The shape parameters consistently decrease both with spatial and temporal aggregation. This means that the smoothing effect due to the spatial and temporal aggregation of rainfall measurement depends on the return period, and is more pronounced for longer return periods. These results relate to the spatial-temporal scales of extreme precipitation, usually analysed using the areal reduction factors, and suggest a non-homogeneity of the scales of rainfall extremes with return period. When using higher spatial and temporal resolutions, it is more probable to observe, in the relatively short archive of radar data, higher extreme events, since they are likely to be more localized in both space and time. This supports previous findings on areal reduction factors derived using radar data (Bacchi and Ranzi, 1996; Durrans, 2002; Allen and De Gaetano, 2005; Lombardo 2006; Overeem et al., 2010). However, it should be noted that other studies reported no clear dependency of the areal reduction factors on return period (Wright et al., 2014).. Magnitude and combination of spatial and temporal effects are shown to depend on the climate, confirming that the spatiotemporal scales of rainfall extremes are highly dependent on the climatic conditions.

### 4.2 Satellite-IDF and radar-IDF

A visual comparison between IDF curves derived from rain gauges and from the co-located radar and satellite pixels is presented in Fig. 4. Corresponding data periods (16 yr, 1 h block) are used over the three climatic regions (Mediterranean–En Hahoresh, semiarid–Beer Sheva and arid–Sedom; Fig. 1) for 1, 6 and 24 h durations. The reported cases, discussed in Marra and Morin (2015), are known to have good radar visibility, with the exception of Sedom that, being farther from the radar and behind the hilly region, can be subject to overshooting of the radar beam. Radar reproduces better the skewness of the IDF curves and radar-IDFs are, with the exception of the arid case, within the rain gauge confidence interval. Note that these results represent the local scale; while interpreting them, one should take into account the different scales of rain gauges (point scale) and remote sensing datasets (~8×8 km$^2$, in this case) and the natural variability that extreme rainfall presents when relatively short records are used (Gires et al., 2014; Peleg et al., 2016).

Examples of IDF maps for 3 h duration from HRC, CHRC and radar products are presented in Fig. 5. It is shown that the spatial variability of IDF values increases with return period, owing to the larger uncertainty associated to longer return periods.

Noteworthy, HRC-IDFs are lower than the corresponding CHRC- and radar-IDFs, while the CHRC-IDFs seem to be larger than radar-IDFs for 2 years return period and comparable for 25 years return period. A quantitative analysis of the differences between HRC/CHRC-IDF maps and radar-IDF maps is provided in the next section.

## 4.3 Comparison between satellite-IDF and radar-IDF

A comprehensive quantitative comparison between radar-IDF and CMORPH-IDF maps is presented in Fig. 6. In general, very similar patterns of CC and NSD are observed for HRC and CHRC while substantially different Bias patterns are observed. This confirms the point made by Mei et al. (2016) that daily gauge adjustment influences the magnitude rather than the space-time organization of annual extremes.

The CC, measuring the spatial correlation of the IDF maps, is as high as ~0.70–0.76 for 2 and 5 years return periods and
decreases when longer return periods are examined, especially for shorter durations (~0.27–0.29 for 1 h, ~0.40–0.55 for 24 h). CHRC has higher CC with radar than HRC. Bias results show that CHRC tends to overestimate while HRC tends to underestimate relative to radar IDFs. For short durations, the Bias for both satellite datasets (gauge adjusted and unadjusted) strongly decreases with increasing return period. As the duration increases, this trend diminishes so that, for longer durations, the Bias is almost consistent for all return periods. This reflects what observed for the shape parameters: the higher shape
parameters estimated from radar for short durations lead the radar IDF curves to be thicker-tailed, i.e. to predict larger intensities for longer return periods with respect to longer durations. Moreover, this suggests that, for sufficiently long durations, it is possible to identify a factor linking radar-IDFs to satellite-IDFs (~0.7 for HRC, ~1.2 for CHRC at 24 h). The NSD patterns are very similar and tend to increase with return period (up to ~0.75 for 25 years, 1 h) and to decrease with duration (~0.55–0.62 for 25 years, 24 h).

Figure 7 and 8 show the metrics calculated between HRC/CHRC-IDF and radar-IDF maps for the three climatic regions and over the sea. The spatial correlation between radar- and CHRC-IDF maps is larger, with respect to HRC-IDF maps, for all the climatic regions and is very high over arid areas/short durations and semiarid areas/long durations. On the contrary, CC between HRC-IDF and radar-IDF maps is very low for long durations (~0.05 for 25 years, 24h), probably due to the very low values measured by CMORPH when no gauge adjustment is applied.

Note that the CC between both HRC/CHRC-IDFs and radar-IDFs is low over the sea, especially for shorter durations, and that the difference becomes less important for longer durations. This is not coming as a surprise since no gauge data is available for the adjustment of satellite or radar data over the sea. As pointed out above, gauge adjustment is only weakly impacting the space-time organization of CMORPH extreme estimates, while it is a crucial step in radar quantitative precipitation estimation. This observation, together with the increased reliability of satellite based estimations over the sea (Kidd and Levizzani, 2011),
suggests that spatial distribution of IDF values indicated by satellite products should be considered more accurate.

Overestimation of CHRC-IDF maps with respect to radar-IDF maps is more marked for Mediterranean and semiarid climates, with a factor ~1–1.6. For durations longer than 3 h, the Bias for Mediterranean and semiarid climates for both HRC and CHRC shows almost no trend with return period, meaning that the trend observed in the general case is due to arid climate and sea

area, where the differences between radar- and HRC/CHRC- shape parameters is larger (Fig. 2). This confirms that radar and HRC/CHRC almost agree in the identification of the skewness of IDF curves over Mediterranean and semiarid climates for durations larger than 3h. NSD over arid and semiarid areas is larger than for Mediterranean climates and sea.

### 4.4 Uncertainty related to the record length

In this section, we present the results of the bootstrap sampling of long rain gauge records used to quantify the uncertainty related to the record length of remote sensing datasets. The uncertainty presented here is the component related to the under-sampling of rainfall climatology due to the use of short data records and is quantified as the 5–95[th] quantile interval of the bootstrap sampling. The uncertainty for two example cases is presented in Fig. 9. The figure reports the 5–95[th] quantile interval of the bootstrap sampling of 10, 15, 20 and 25 years of data out of the whole record of the three cases shown in Fig. 4. As

expected, uncertainties become important when the record length is similar, or smaller, than the estimated return period. Arid climates are characterized by larger uncertainties, especially when short records and short durations are examined; this is probably due to the low number of rain events per year, and to the thicker-tail characteristic of arid IDF curves. Short records are shown to be more likely overestimating, rather than underestimating, the IDF values.

Figure 10 shows the relative uncertainty (width of the 5–95[th] quantile interval of the 999 bootstrap sampling repetitions

normalized over the long record rain gauge-IDF value) as a function of the ratio between return period and record length. Uncertainties for 1 h and 3 h durations are comparable, with the uncertainty for 3 h duration being smaller. This suggests that time aggregation potentially decreases part of the issues related to the use of short records. Uncertainty is larger for return periods longer than the record length, especially for short durations and drier climates. For the Mediterranean climate, the uncertainty is generally lower than 50% when the return period is shorter than the record length and reaches up to ~125% for

return periods more than twice the data record length. In the semiarid climate, the uncertainty is shown to be larger, exceeding 100% (30–100%) even for return periods comparable to the data record length and exceeding 200% for return periods more than twice the data record length. In the arid climate, the uncertainty is even larger, reaching 150% for return periods equal to the data record length and exceeding 250% for return periods more than twice the data record length.

### 5 Conclusions

This study compared the use of rainfall estimates from a ground based C-band weather radar and from a high-resolution satellite precipitation product, CMORPH (HRC) and its gauge-adjusted version (CHRC), for the identification of Intensity–Duration–Frequency curves. IDF curves were computed using the above products over the Eastern Mediterranean (Mediterranean, semiarid and arid climates and over the sea) and the uncertainties due to the limited record length of the remote sensing datasets were quantified basing on long records of rain gauge measurements. Our findings can be summarized as

follows:

1. The shape parameters of the generalized extreme values distribution, as derived from radar, HRC and CHRC, are mostly greater than zero; drier climates are characterized by higher shape parameters, suggesting that thicker-tail distributions better describe rainfall extremes of drier areas. In general, the shape parameters derived from radar are higher than the ones from CMORPH, especially for arid climate and over the sea.

2. The shape parameter tends to decrease when rainfall estimates are aggregated in space and/or time. The effect is related to a non-homogeneity of spatial and temporal scales of rainfall extremes with return period. This non-homogeneity depends on the climatic conditions.

3. The spatial correlation coefficient between corresponding radar-IDF and HRC/CHRC-IDF maps is between 0.70 and 0.76 for short return periods, but decreases with increase in return period, especially for short durations. In general, for both HRC and CHRC the correlation is higher in arid climate for durations up to 3 h and in Mediterranean and semiarid climates for longer durations (6–24 h). Low correlations are observed over the sea.

4. HRC-IDFs and CHRC-IDFs are, respectively, lower and higher than radar-IDFs. In both cases the observed Bias decreases with return period, especially for short durations and arid climate. For longer durations and Mediterranean/semiarid climates, the decreasing trend almost disappears so that the Bias can be considered independent from the return period (~0.7 for HRC, ~1.2 for CHRC for 24 h).

5. Comparison of HRC-IDF and CHRC-IDF against radar-IDF show consistent patterns of correlation and dispersion, and different biases. This means that gauge-adjustment influences the magnitude rather than the space-time organization of annual extremes and suggests that HRC-IDF can potentially be used to estimate the frequencies of CMORPH estimates in near real time early warning systems.

6. The uncertainty related to the use of short records becomes important when the record length is shorter or comparable to the examined return period. This is particularly true for drier climates and shorter durations, with potential uncertainty of ~50%, ~100% and ~150% for return periods comparable to the record length in Mediterranean, semiarid and arid climates, respectively.

Rainfall frequency analysis by means of remote sensing rainfall estimates remains a challenging task, especially when dry climates are explored. Nevertheless, the agreement between IDF curves derived from different sensors on Mediterranean and, to a good extent, semiarid climates, demonstrates their potential for the description of small scale spatial patterns of IDF curves and instils confidence on their quantitative use for ungauged areas of the Earth. Spatial and temporal aggregation of rainfall information represent viable ways to take advantage of remote sensing datasets and decrease the uncertainties related to the derived IDF curves. In particular, remote sensed rainfall archives can provide important information when 2–10 years return periods and 12–24 h durations are requested, scales that are relevant for both flood risk management (e.g. issuing of warning) and hydrological design (e.g. sewer systems design, large scale drainage planning).

## Appendix A. GEV distribution

The GEV cumulative distribution function can be written as (Coles, 2001):

$$\begin{cases} F(I; \mu, \sigma, \kappa) = \exp\left\{ -\left[ 1 + \frac{\kappa}{\sigma}(I - \mu) \right]^{-\frac{1}{\kappa}} \right\} & \text{for } \kappa \neq 0 \\ F(I; \mu, \sigma, \kappa) = \exp\left\{ -\exp\left[ -\frac{1}{\sigma}(I - \mu) \right] \right\} & \text{for } \kappa = 0 \end{cases} \tag{A.1}$$

where $I$ is the average intensity over a given duration and $\mu$, $\sigma$ and $\kappa$ are respectively the location, scale and shape parameters

of the distribution. The shape parameter is directly related to the tail of the extreme values distribution: low (high) shape parameters are related to lower (higher) probability of having large extremes. When the shape parameter is lower than 0 (type III distribution), an upper limit to the extreme values is expected. When the shape parameter is greater (type II) or equal (type I distribution) to 0, no upper limit to the values is expected. In particular, when the shape parameter is greater than 0.5, the standard deviation of the distribution is infinite and when it is greater than 1, the mean of the distribution is infinite. Under

hypotheses on the regularity of the tail of the distribution, the Fisher-Tippet theorem demonstrates that GEV distribution is the only possible limit distribution for the extreme values of independent and identically distributed random variables.

## Appendix B. Comparison metrics

*Correlation Coefficient* (CC) measures the spatial correlation of the derived maps. It is calculated as:

$$CC = \frac{\text{Cov}(\boldsymbol{r}, \boldsymbol{s})}{\text{StdDev}(\boldsymbol{r}) \cdot \text{StdDev}(\boldsymbol{s})} \tag{B.1}$$

where $r$ and $s$ refer to the radar-IDF maps and satellite-IDF maps (HRC or CHRC), respectively. It ranges between -1 and 1 with positive (negative) values indicating positive (negative) correlation.

Multiplicative *Bias* measures the mean quantitative agreement of two maps. It is calculated as:

$$\text{Bias} = \frac{\frac{1}{N}\Sigma_i^N s_i}{\frac{1}{N}\Sigma_i^N r_i} \tag{B.2}$$

where $r_i$ and $s_i$ are the radar-IDF and satellite-IDF estimates respectively, and $i$ is the index of a pixel within the $N$ pixels

composing the maps. It may assume any positive value with 1 indicating perfect mean quantitative agreement, value higher (lower) than 1 indicating that the mean satellite estimate is higher (lower) than the mean radar estimate.

*Normalized Standard Difference* (NSD) measures the standard deviation of the residuals of the normalized maps and is calculated as:

$$\text{NSD} = \sqrt{\frac{1}{N}\Sigma_i^N \left( \frac{s_i}{\hat{s}} - \frac{r_i}{\hat{r}} \right)^2} \tag{B.3}$$

where $\hat{s}$ and $\hat{r}$ are the mean values of satellite and radar maps. NSD may assume any value $\geq 0$.

## Acknowledgments

Radar data were provided by E.M.S. (Mekorot Company) and rain gauge data were provided by the Israel Meteorological Service. The study was partially funded by the Lady Davis Fellowship Trust [project: RainFreq], by the Israel Science Foundation [grant no. 1007/15], by the PALEX DFG project, and by NSF-BSF grant [BSF 2016953]. This work is a contribution to the HyMeX program. We thank the three anonymous reviewers for contributing improving the paper and Luca Panziera for the helpful discussions. The original and gauge-adjusted CMORPH precipitation products are downloadable from ftp://ftp.cpc.ncep.noaa.gov/precip/CMORPH_V1.0/RAW/8km-30min/ and ftp://ftp.cpc.ncep.noaa.gov/precip/CMORPH_V1.0/CRT/8km-30min/, respectively (accessed November 2015).

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

**Table 1: Koppen-Geiger classification (Peel et al., 2007; Srebro and Stoffer., 2011) and number of pixels analyzed for each climatic region according to the Koppen-Geiger classification. Note that: MAT = Mean Annual Temperature; MAP = Mean Annual Precipitation; $T_{hot}$ = Temperature of the hottest month; $T_{cold}$ = Temperature of the coldest month; $P_{ds}$ = Precipitation of the driest month in summer; $P_{ww}$ = Precipitation of the wettest month in winter.**

| Climatic region | Koppen-Geiger definition | Koppen-Geiger criteria | Number of pixels (fraction of the total) |
|---|---|---|---|
| *Mediterranean* | Temperate, Dry summer, Hot summer (Csa) | $T_{hot} \geq 22°C$ <br> $0°C < T_{cold} < 18°C$ <br> $P_{ds} < 40$ mm <br> $P_{ds} < P_{ww} / 3$ | 318 (20.1%) |
| *Semiarid* | Arid, steppe, hot (BSh) | $10 \times MAT \leq MAP \leq 20 \times MAT$ <br> $MAT \geq 18°C$ | 404 (25.5%) |
| *Arid* | Arid, desert, hot (BWh) | $MAP < 10 \times MAT$ <br> $MAT \geq 18°C$ | 225 (14.2%) |
| *Sea* | - | - | 635 (40.1%) |

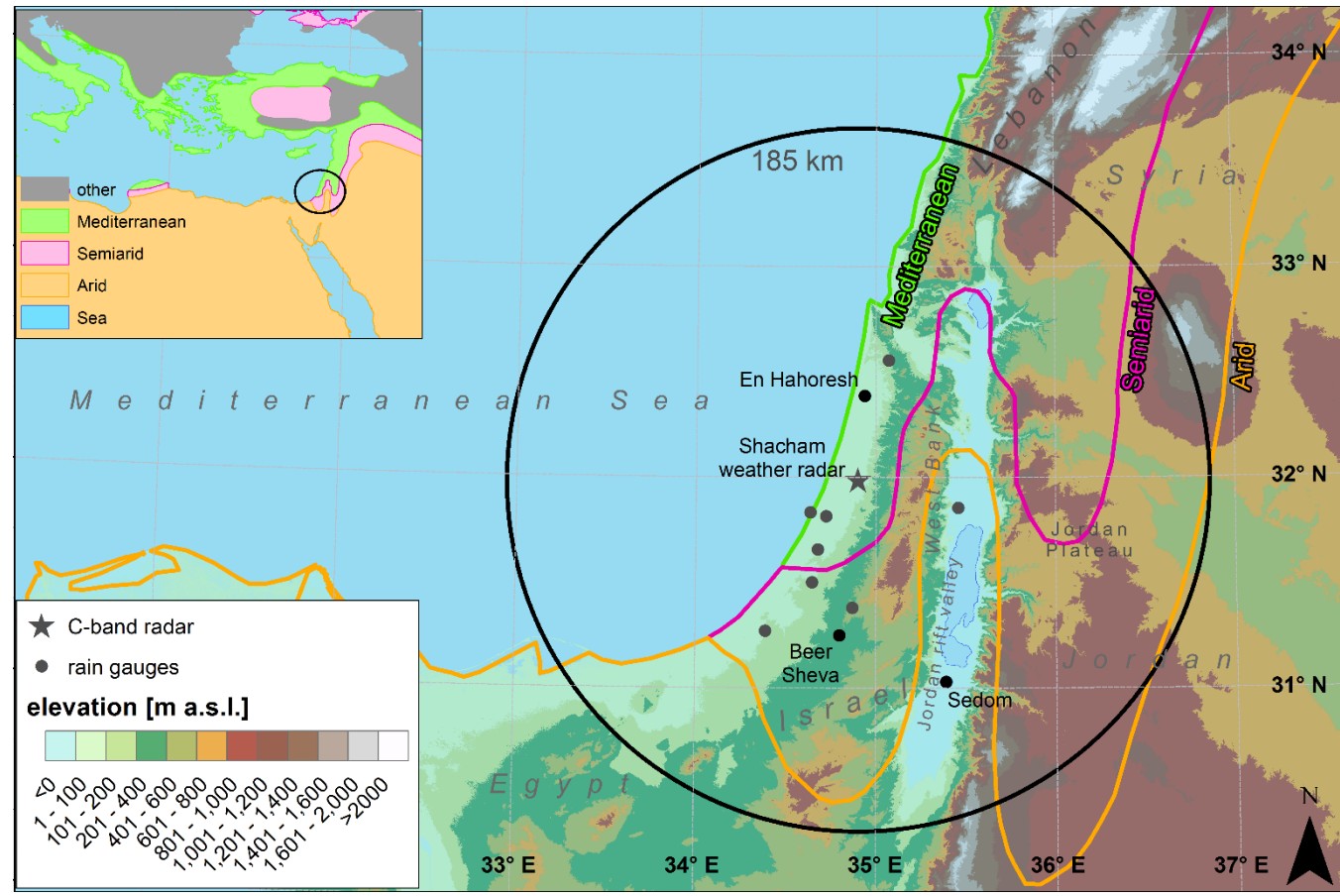

**Figure 1: Map of the study area showing terrain elevation, climatic classification from Shahar et al. (2008), location of the radar and rain gauges used in the study. The small map shows the location of the study area within the Eastern Mediterranean.**

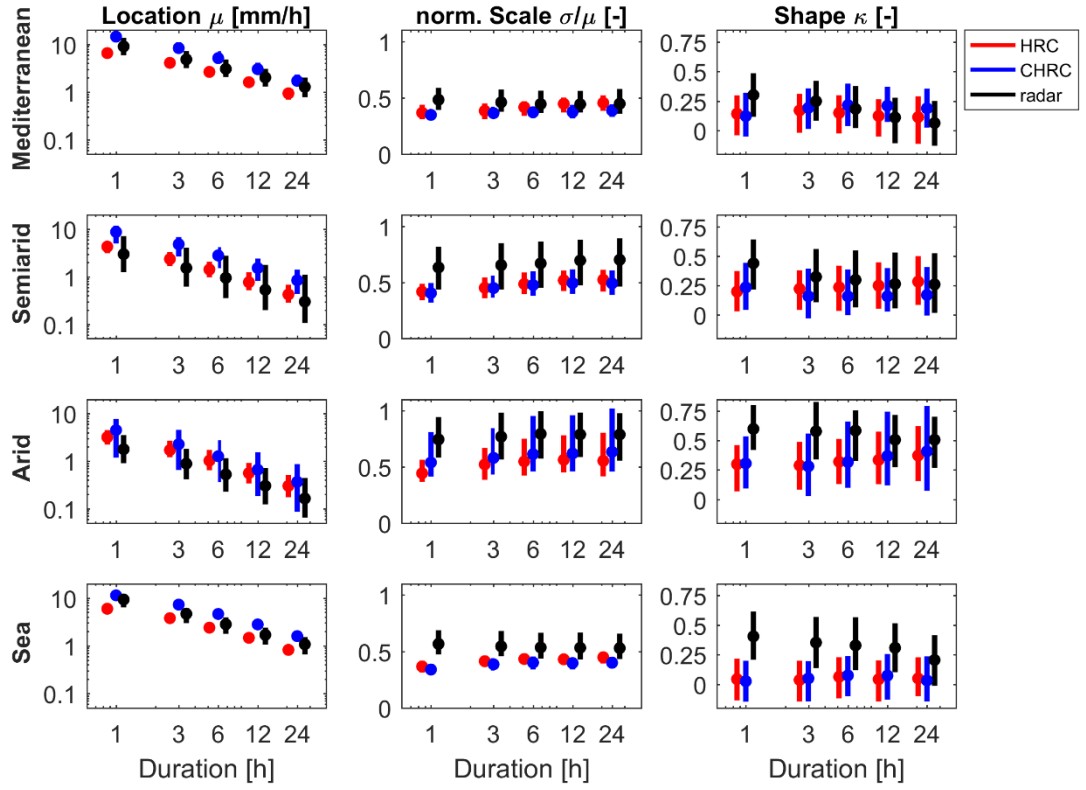

Figure 2: Distribution (median and 25th-75th quantiles) of the GEV parameters derived from satellite (HRC and CHRC) and radar datasets. Note that scale parameters are normalized over the corresponding location parameters. The parameters for different products are represented around the corresponding duration so the logarithmic scale in x-axes should be interpreted accordingly.

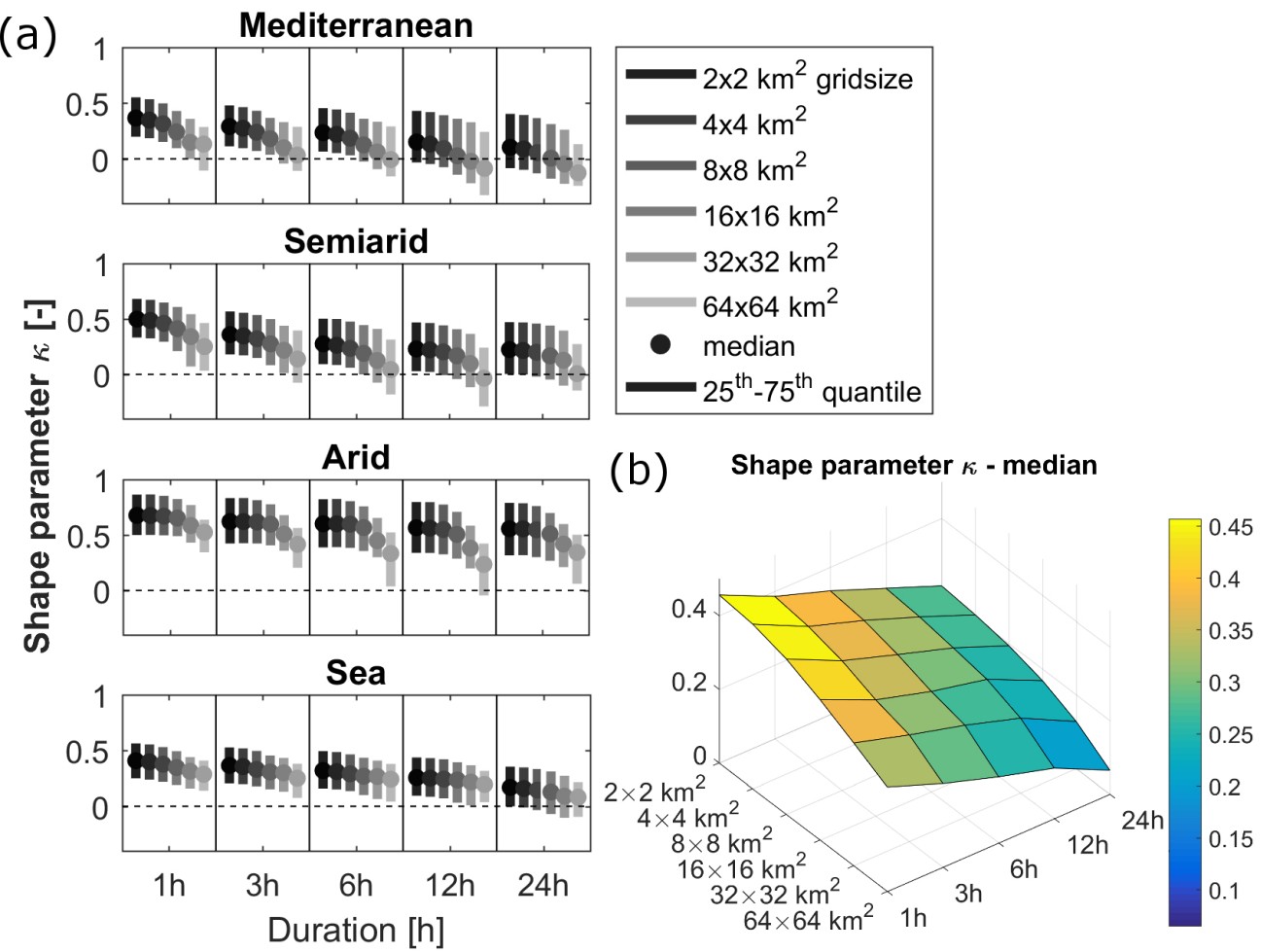

**Figure 3: (a) Distribution (median and 25th -75th quantiles) of the shape parameters derived aggregating radar estimates on grid sizes increasing from 2×2 to 64×64 km2. (b) Median of the shape parameters (all climates) derived aggregating radar estimates as a function of the spatial and of the temporal aggregation scales.**

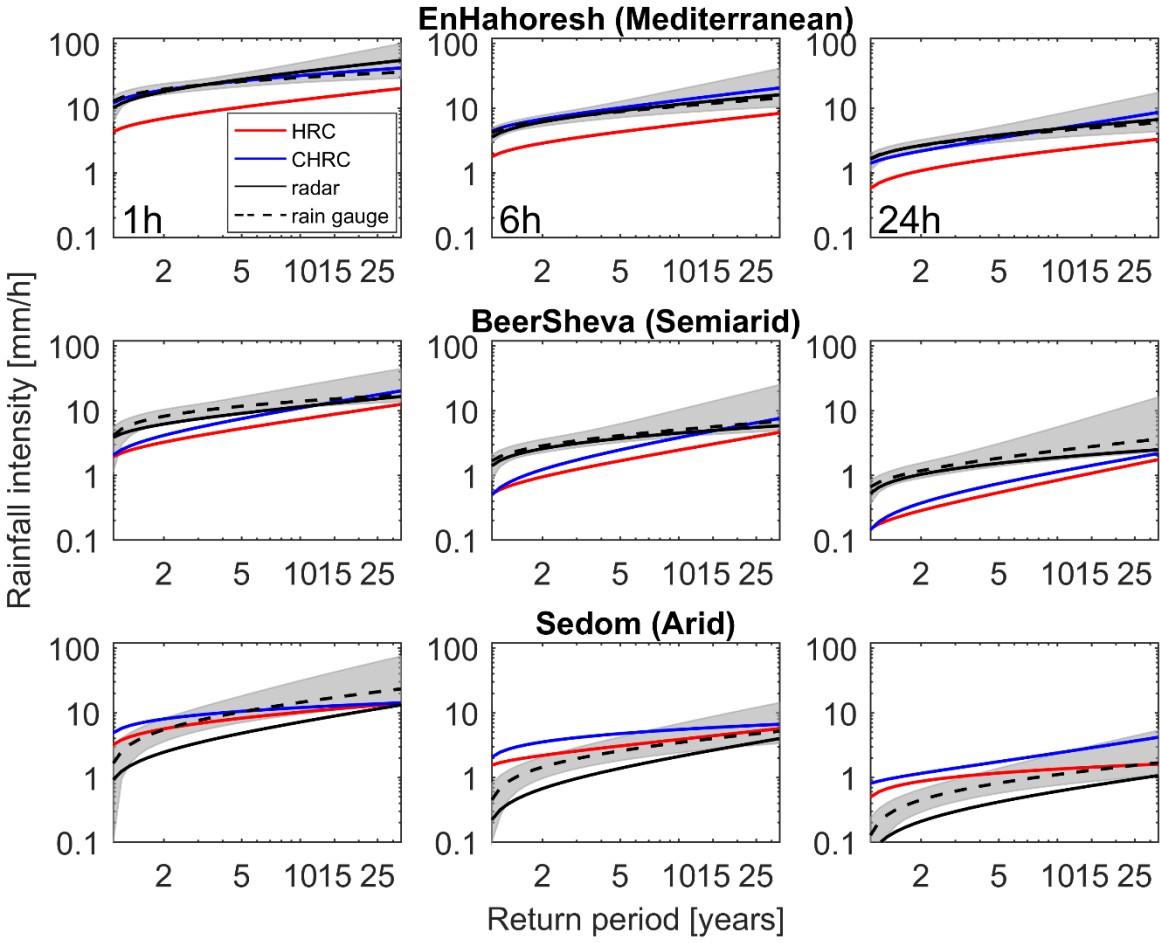

**Figure 4: Visual comparison of the annual maxima and of the IDF curves derived for the examined 16 years (1998-2013) from HRC (red), CHRC (blue), radar (solid black) and rain gauge (dashed black). The shaded area represents the 95 % confidence interval of the rain gauge IDF. Three example locations in Mediterranean (En Hahoresh), semiarid (Beer Sheva) and arid (Sedom) climates and for 1, 6 and 24 h durations are shown (see gauge location in Fig. 1).**

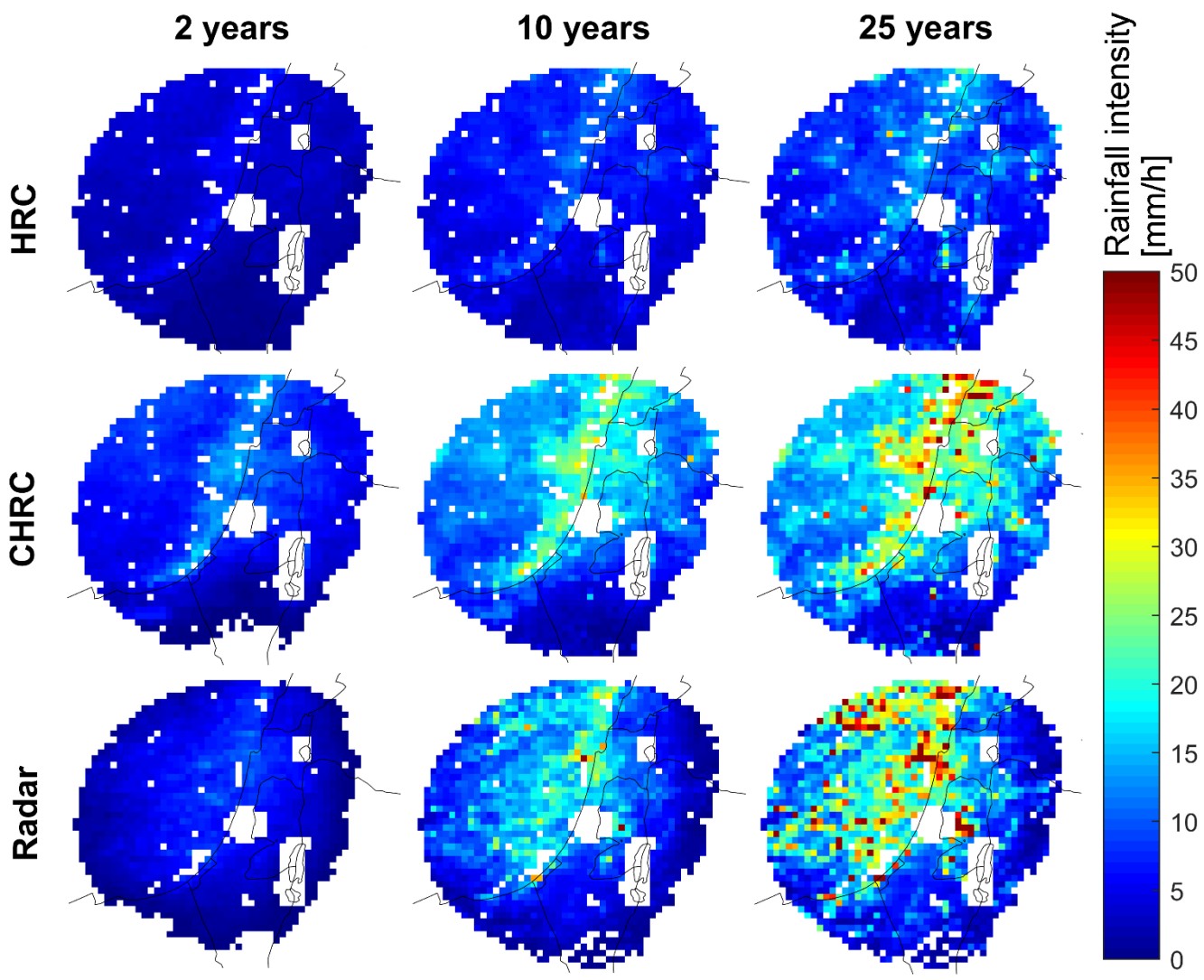

**Figure 5: Example of IDF maps for 3 h duration from HRC, CHRC and radar for 2, 10 and 25 years return periods. Only the pixels included in the analyses are shown.**

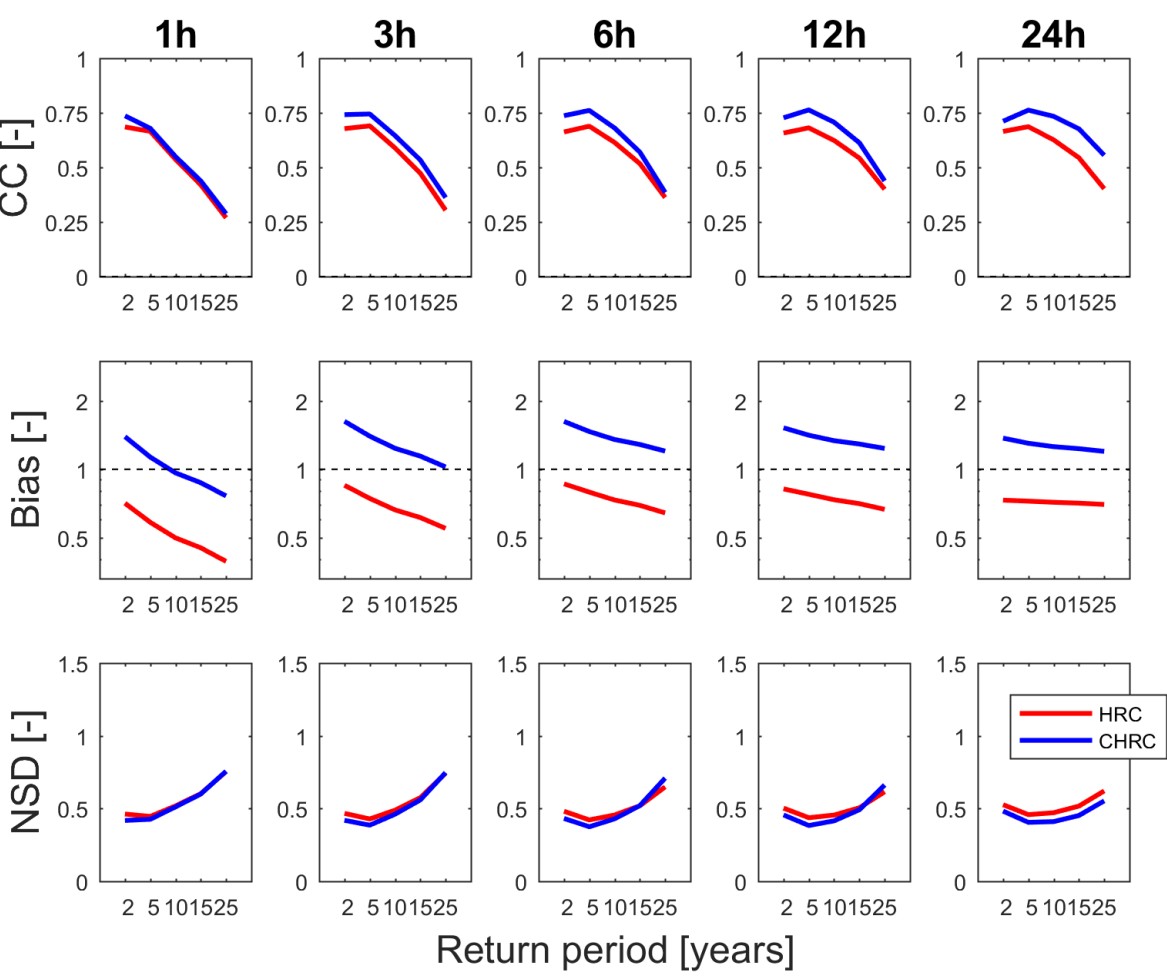

**Figure 6: Comparison of IDF values between HRC and radar (red) and CHRC and radar (blue). The first row of panels shows the CC for different durations, the second row the Bias and the third row the NSD.**

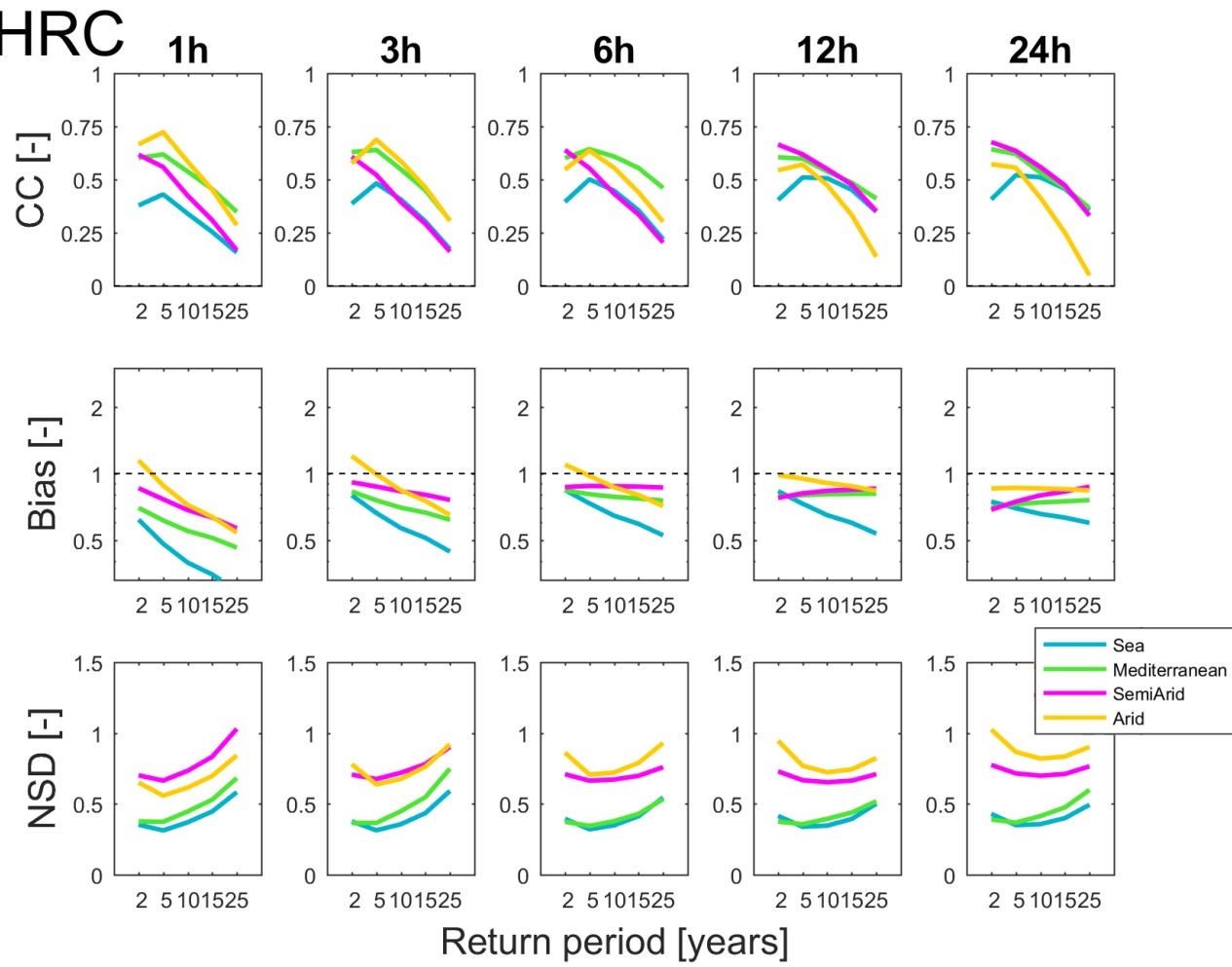

**Figure 7: Comparison of IDF values between HRC and radar for different climatic regions. Blue lines represent sea areas while green, pink and orange lines represent Mediterranean, semiarid and arid climates respectively. The first row of panels shows the CC for different durations, the second row the Bias and the third row the NSD.**

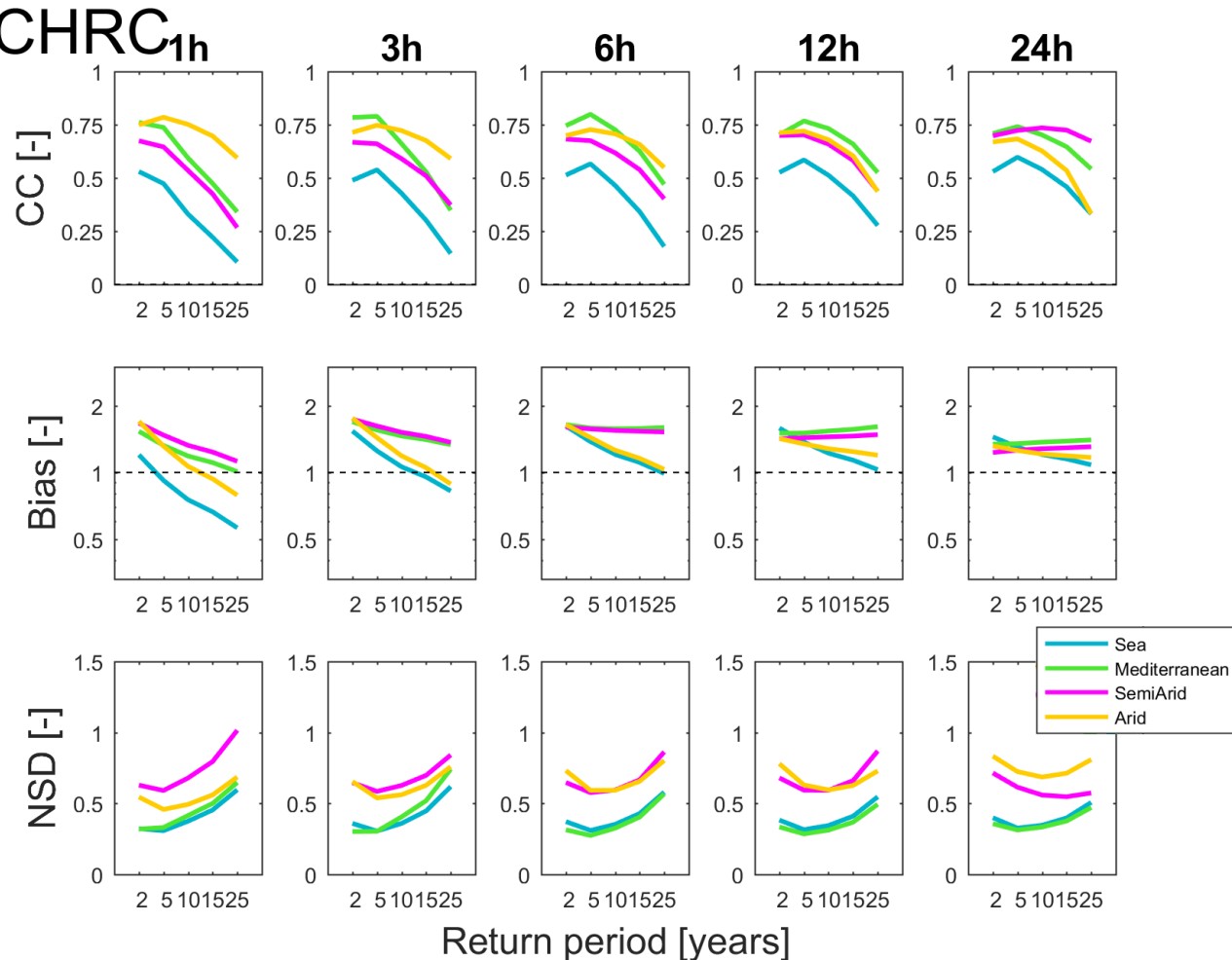

**Figure 8: Comparison of IDF values between CHRC and radar for different climatic regions. Blue lines represent sea areas while green, pink and orange lines represent Mediterranean, semiarid and arid climates respectively. The first row of panels shows the CC for different durations, the second row the Bias and the third row the NSD.**

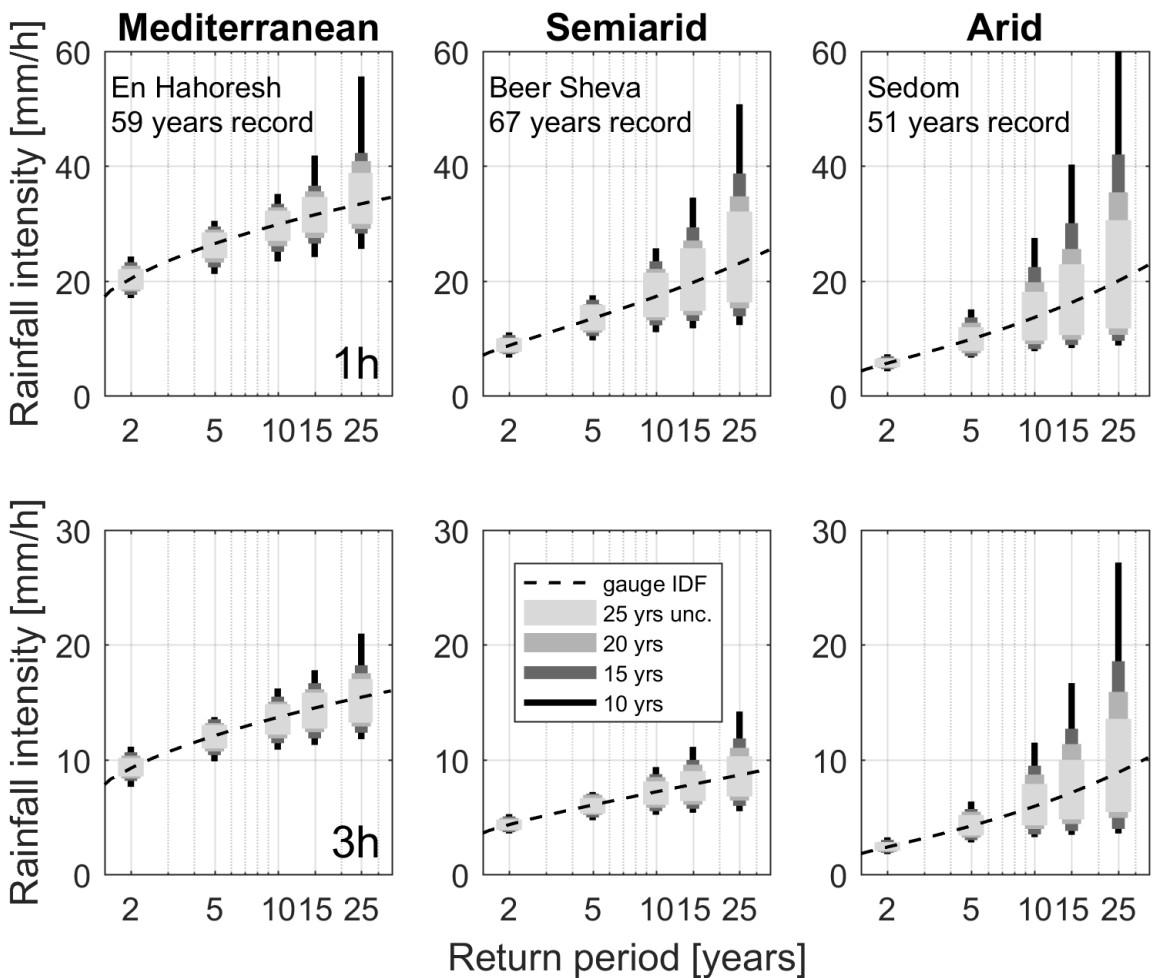

Figure 9: Uncertainty related to the record length for 1 and 3 h durations for three example cases in Mediterranean (En Hahoresh), semiarid (Beer Sheva) and arid (Sedom) climates. The dashed lines show the IDF curves from the full records (59, 67 and 51 years, respectively) and the vertical bars show the width of the 5-95th quantile interval of the 999 bootstrap sampling repetitions for record lengths of 10, 15, 20 and 25 years. Width and light of the colour of the bars increase with the record length.

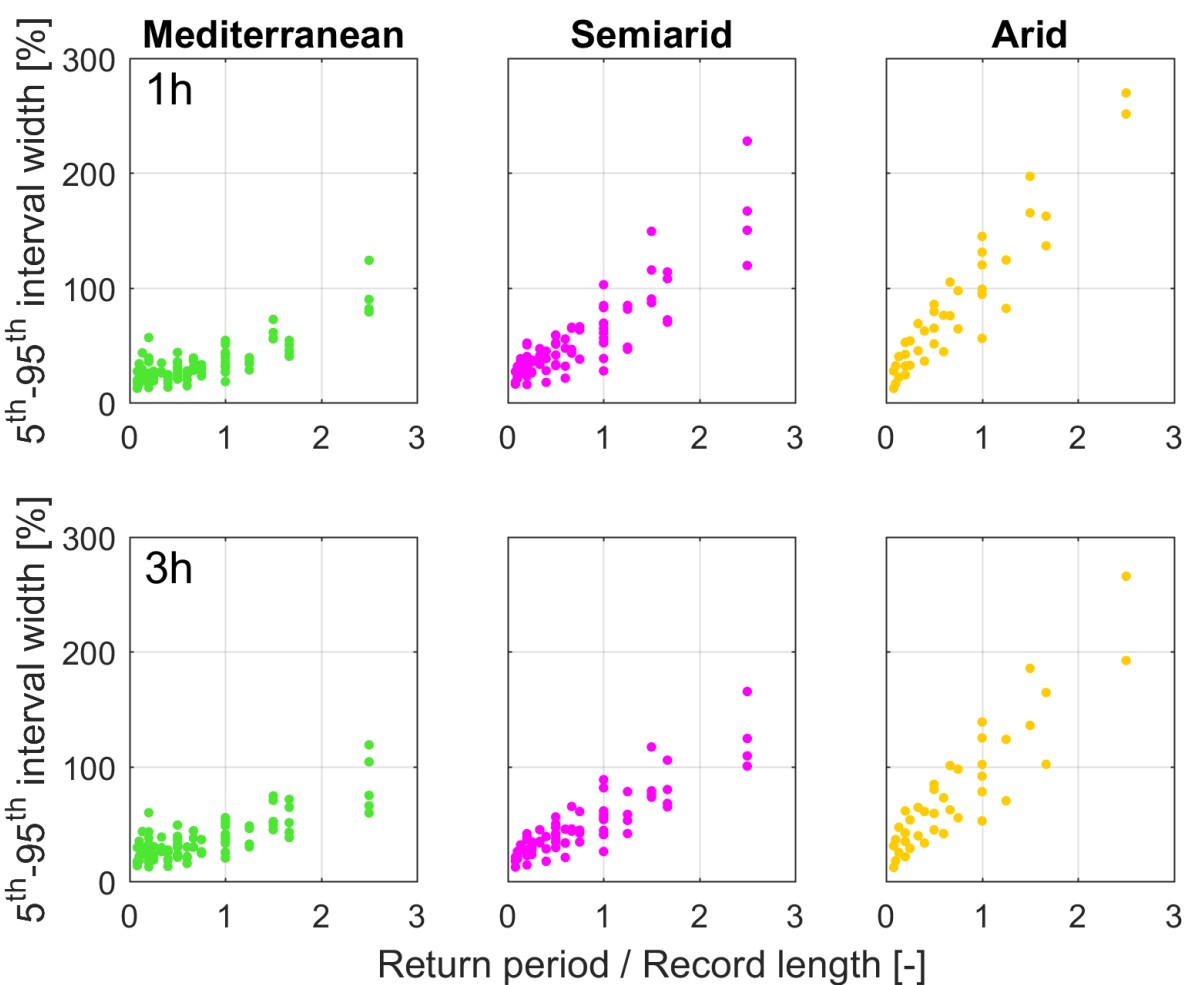

**Figure 10: Relative uncertainty (width of the 5-95[th] quantile interval of the 999 bootstrap sampling repetitions normalized over the rain gauge-IDF value) plotted against the ratio between the estimated return period and the record length. The first row of panels shows the results for 1 h duration, the second row for 3 h. Columns of panels show results for each climatic region.**

