# Peer review of "Intensity-Duration-Frequency curves from remote sensing rainfall estimates: comparing satellite and weather radar over the Eastern Mediterranean"

_Hydrology and Earth System Sciences, 2016_

## Referee Comment (RC1) · Anonymous Referee #1 · 11 Dec 2016

Dear Editor,

Manuscript entitled "Comparing Intensity–Duration–Frequency (IDF) curves derived from CMORPH and radar rainfall estimates over the Eastern Mediterranean" has been reviewed. In the paper, the authors compared IDF curves from radar and CMORPH in different climatic regions. They found that radar shows thicker tail distributions than CMORPH across the Eastern Mediterranean. The authors do not present a new tool. However, the manuscript is well organized. I divided my review into general comments and specific ones.

[Figure]

General comments:

1) The Koppen-Geiger classification used are not mentioned or described in section 2.1.

2) The lack of precision on the applied data leads to misunderstand some results: if the authors are using adjusted CMORPH gauge, what's the reason for analysis the unadjusted CMORPH data (in the same spatiotemporal resolution). I think there is something missing to clarify the methodology.

3) How is spatial (upscaled) data thought to affect precipitation (mainly IDF for return periods)? It is not clear to the readers without background knowledge of the appropriate upscaling method over the study area. Does an adopted upscaling approach perform better than some other techniques (this may not be necessary, but it should be discussed at least)?

4) Any thoughts on the reason why upscaling data is only found significant on the shape parameter (Section 4.1.1)? Maybe I am missing something in this part, but that is my understanding.

5) On the same subject, what is the CMORPH spatial upscaling (downscaling) impacts on precipitation?

6) Why overload figures with features with the insufficient discussion in the text (e.g., Lack of enough discussion for location and scale parameters or for a different time interval in Fig 4)?

The figures axis could be improved if the authors want to make it clear to the reader. Also, I suggest the same size intervals in Y-axis (e.g., Fig 2&4).

7) Is there any physical link that could impacts on estimations due to different spatial resolution?

8) Some unnecessary materials: in my opinion, it is completely unnecessary to have

Figure 9, when it is already well-known that the uncertainties increase for longer return period.

9) Finally, I would like to draw your attention to a somewhat old, but still relevant, review on methods to IDF and DDF curves and their uncertainties that might have some valuable lessons in it: Aart Overeem, Adri Buishand and Iwan Holleman, Journal of Hydrology, 2008, 348, pp 124-134

Specific Comments:

Page 5- Line 15:

- I would expect to introduce parameters and explain a little bit how they were computed.

- It is not clear which method used for estimating the GEV distribution parameters by maximum likelihood.

- The bootstrap was applied to assess this uncertainty of record length. This method considers only the uncertainty for the return periods (it's my understanding). What about the uncertainty of the estimation of the GEV parameters (i.e. sampling errors)?

- The assumptions regarding the convenience of the GEV to represent the AMS distribution through the Fisher-Tippett theorem, as well as calculation procedures, would be useful to present in the Appendix.

Page6- Line 12:

- Figure 2: Estimates of the GEV parameters could be present by Box-Whisker plot that obtained from 999 bootstrap samples, as well for 25th and 75th percentiles of the bootstrap samples. It is also suggested to present the parameter variation against the time by a linear representation that could help the readers to grasp the message from the figures.

- How are different climatic regions thought to affect precipitation (location and scale

parameters)? It should be discussed at least (regarding figure 2).

Page 6-

- The location and scale parameters are not addressed in details. It is not clear to the reader without background knowledge of the GEV df.

Page 8-

- I am totally lost. What's the link between section 4.4 with previous sections?

Page 12- Line 13:

- Sorry if I missed this, how is the coverage of upscaled radar and CMORPH pixels? Are they exactly the same? Please, explicitly specify their spatial coverage.

Page 18-

- What's the dashed line in Fig 2 for shape parameter?

Page 20-

- Appendix A. The legend and description in caption do not show similar things (check).

Page 25-

- Why Fig 9 presented? It is well known that confidence in a return level decreases rapidly when the period is more than about two times the length of the original data set. Fig 9 could be removed.

---

## Referee Comment (RC2) · Anonymous Referee #2 · 15 Dec 2016

The manuscript "Comparing Intensity–Duration–Frequency curves derived from CMORPH and radar rainfall estimates over the Eastern Mediterranean" represents a good contribution to the assessment of satellite-based rainfall estimations in areas where their impact could be potentially relevant. The authors have done a thorough analysis over a sufficient amount of time and thus the results are significant.

The manuscript certainly fits the journal scope and is suggested for publication after the following points are considered by the authors:

1) The reason why both versions (gauged and ungauged) of CMORPH are used is not

sufficiently explained and detailed in the text.

2) Did the authors check how many of the gauges are used in the production of the gauge-adjusted version of CMORPH in the area? This could have an impact in the analysis, maybe small. However, a couple of words on the subject need to be included.

3) While it is true that gauges are not available over the sea (!) the authors should also spend a few words on the fact that normally satellite-based estimations are far better over the sea surface. This partially contradicts their results. Just pay attention to this important fact. See, for example: Kidd, C., and V. Levizzani, 2011: Status of satellite precipitation retrievals. Hydrol. Earth Syst. Sci., 15, 1109-1116 for a discussion on the various methods of rain estimation from satellite.

4) A fine combing of the English is suggested since many imperfections are detected throughout the text.

Minor point: The caption of Fig. 4 does not match the real colors used in the figure. It appears that the authors have used two different version prior and after a change they did while writing.

---

## Author Comment (AC1) · 12 Jan 2017

**Manuscript reference number: hess-2016-597 - Response to Anonymous referee #1**

We would like to thank the referee for the helpful comments. Below we provide our response to the reviewer's comments and describe the modifications made to address them. In black font we include the reviewer's comments, and in blue is our response.

Manuscript entitled "Comparing Intensity–Duration–Frequency (IDF) curves derived from CMORPH and radar rainfall estimates over the Eastern Mediterranean" has been reviewed. In the paper, the authors compared IDF curves from radar and CMORPH in different climatic regions. They found that radar shows thicker tail distributions than CMORPH across the Eastern Mediterranean. The authors do not present a new tool. However, the manuscript is well organized. I divided my review into general comments and specific ones.

We would like to thank the referee for the detailed review.

This paper aims at advancing knowledge on the use of remote sensed data for rainfall frequency analysis in the form of IDF curves. The intent of this study is not to present a new "tool" for IDF estimation, but explore the use of remotely sensed precipitation from high-resolution satellite precipitation products vs. the more standard gauge-adjusted ground radar-rainfall fields. The use of standard tools for IDF estimation is crucial to avoid methodological issues masking out the results. To our knowledge this is the first study in which at-site IDF curves derived from different gridded datasets (from completely different sensors; with and without gauge-based adjustment) are compared. This approach strongly reduces the (spatial) scale issues related to the comparison of point and areal estimates, that hampered the research so far and potentially opens the way to apply remote sensing for deriving IDFs in data poor areas.

In order to make this aspect clearer, we modified the title of the manuscript to: "Intensity-Duration-Frequency curves from remote sensing rainfall estimates: comparing satellite and weather radar over the Eastern Mediterranean" and revised introduction as follows: "Since quantitatively accurate information is essential for design applications and the derivation of IDF curves based on historical records does not require short latency in the data, these studies made use of gauge-adjusted products and assessed the accuracy of the IDF curves derived from remote sensing datasets using rain gauge curves as a reference. However, this approach neglected two important aspects. First, early warning systems, e.g. for flash floods (Borga et al., 2011; Villarini et al., 2010; Borga et al., 2014), landslides/debris flows (Tiranti et al. 2014; Borga et al., 2014; Segoni et al., 2015) or heavy rain (Panziera et al. 2016), need to operate in real-time and rely on short-latency remote sensed measurements. In these situations, calculating the frequency of near real-time estimates using IDF curves derived from gauge-adjusted data could provide misleading results. It is therefore useful to analyse the characteristics of IDF curves derived from non-adjusted rainfall data, which are expected to represent the frequencies of near real time estimates. Second, areal IDFs provided by remote sensing instruments are expected to differ from point IDFs (Peleg et al., 2016a), and the use of different records (i.e. different samples of the climate) introduces further differences. No exact match between remote sensing and rain gauge IDFs should be expected a priori [...] To the authors' knowledge this is the first study in which at-site IDF curves derived from different gridded remote sensing datasets are compared". [this will be recalled in the answer to reviewer's second comment].

**General comments:**

1) The Koppen-Geiger classification used are not mentioned or described in section 2.1. Thank you for pointing this out. We introduced the classification in section 2.1 as: *"Three climatic regions, Mediterranean, semiarid and arid, can be identified in the area, corresponding to the Csa, BSh*

**and BWh Koppen-Geiger definitions, respectively (Peel et al., 2007). The criteria used to define the classes are reported in Table 1. In this study, we follow the classification by Srebro et al. (2011)."**

2) The lack of precision on the applied data leads to misunderstand some results: if the authors are using adjusted CMORPH gauge, what's the reason for analysis the unadjusted CMORPH data (in the same spatiotemporal resolution). I think there is something missing to clarify the methodology.

Thank you for raising this important issue – also shared by reviewer 2. As underlined in the general response, our introduction failed to emphasize the importance of using both the gauge-adjusted and unadjusted versions of the satellite product (CMORPH, is this case). The gauge adjustment is expected to improve satellite estimates on average at the cost of having this improved product with several days of latency, which does not allow the use in real time applications. Furthermore, we would like to mention that there are regions on earth where there are no ground based sensors to adjust the satellite precipitation datasets. Hence, the natural question by the reviewers: 'why are you using the non-adjusted product in an application that is based on historical data rather than real time estimates?' The answer lies in the different requirements of the applications that make use of IDF curves: hydrologic design needs optimal quantitative accuracy, so evaluating unadjusted data is important for demonstrating uses of satellite precipitation in ungauged areas; early warning systems need to correctly identify the frequency of near real-time estimates. We updated text in the introduction to better motivate our study as mentioned above in the general response to the reviewer, and we recalled the concept in the conclusions, adding a new point: "Comparison of HRC-IDF and CHRC-IDF against radar-IDF show consistent patterns of correlation and dispersion, and different biases. This means that gauge-adjustment influences the magnitude rather than the space-time organization of annual extremes and suggests that HRC-IDF can potentially be used to estimate the frequencies of CMORPH estimates in near real time early warning systems."

In addition to this, it is interesting to note that the performance of gauge-adjustment on extremes is affected by the different measurement scales of remote sensing instruments and rain gauges. Moreover, the impact of gauge-adjustment over ungauged areas is rarely quantified, especially for extremes and potentially introduces non homogeneity in space due to different gauge density data.

**[We group here two comments dealing with the same topic].**

3) How is spatial (upscaled) data thought to affect precipitation (mainly IDF for return periods)? It is not clear to the readers without background knowledge of the appropriate upscaling method over the study area. Does an adopted upscaling approach perform better than some other techniques (this may not be necessary, but it should be discussed at least)?

7) Is there any physical link that could impacts on estimations due to different spatial resolution? Thank you for raising these questions. The relation between point and areal extreme rainfall is the subject of research for a long time. As explained in the introduction, the reason for this interest is that the classic approach derives IDF curves from point data (gauges) while for many applications the areal information (e.g. in a catchment scale) is required. The classic tool for converting point to areal information is the Areal Reduction Factor (ARF) method, already mentioned in the manuscript. We think that discussion on ARF methods is beyond the scope of this paper since it focuses on areal gridded information from remote sensing datasets: no upscaling method is used, no ARF is provided. However, we added text in the introduction to improve this aspect, also providing a useful reference: "In principle, areal reduction factors may depend on a number of factors, such as geographic location, characteristics of the examined catchment, analysed duration and period, season, meteorological conditions, and others (Svensson and Jones, 2010). Their derivation is thus hampered by many sources of uncertainty."

4) Any thoughts on the reason why upscaling data is only found significant on the shape parameter (Section 4.1.1)? Maybe I am missing something in this part, but that is my understanding

Upscaling data significantly affects the location and scale parameters ("The location and scale parameters, as expected, consistently decrease as the spatial and temporal aggregation scales are increased and are not reported here"). The impact of spatial aggregation on mean and dispersion of the GEV distribution is related to the spatial smoothing of the extreme rainfall fields. This causes a decrease of the location and scale parameters. As in the case of ARFs, this decrease depends on a number of factors, which are difficult to isolate, and whose investigation falls out of the scope of this study. Conversely, the impact of spatial aggregation on the shape parameter can be easily assessed with the data in our possession and provides useful insights to the community. We revised the sentence to make it clearer: "The location and scale parameters consistently decreased as the spatial and temporal aggregation scales increased. This is an expected effect, caused by the smoothing of rainfall fields operated by the spatial averaging, therefore results are not reported in this paper. Conversely, it is interesting to analyse the shape parameter.".

In general, we believe that improving our knowledge on the interpretation of areal precipitation information is key for the future research in the field.

5) On the same subject, what is the CMORPH spatial upscaling (downscaling) impacts on precipitation? We thank the reviewer for the question. The rationale for using weather radar data, rather than CMORPH data, for this analysis is twofold: (a) CMORPH estimates, even if provided at ~8 km resolution, are derived from coarser resolution PMW retrievals, so the relatively-small scale spatial structures are difficult to be captured with this product; (b) radar data allows to explore a broader range of scales. Following the suggestion by the reviewer, we repeated the same analyses (starting from 8 km gridsize)

using the CHRC data. The decrease of shape parameters with gridsize is less marked, but confirmed. Conversely, shape parameters slightly increased with duration: they are almost uniform for 8 km gridsize (between 0.14-0.16) and increase from  $\sim$ 0.02 (1h) to  $\sim$ 0.14-0.15 (12h-24h) for 64 km. The variations are less marked than in the radar case, since the shape parameters from CMORPH are already low. We however think that CHRC provides less accurate information, due to the point (a) above.

6) Why overload figures with features with the insufficient discussion in the text (e.g., Lack of enough discussion for location and scale parameters or for a different time interval in Fig 4)?

We thank the reviewer for this comment. We revised Fig. 2 showing the distribution of the scale parameters normalized over the corresponding location parameters (dispersion normalized over the mean). The text in section 4.1 is subsequently revised as follows: "Location parameters from HRC (CHRC) estimates are smaller (larger) than the ones from radar over Mediterranean climate and over the sea, meaning that extreme values from HRC (CHRC) are in general lower (higher) than radar extreme values while in semiarid and arid climates HRC and CHRC generally identify higher parameters than the radar (i.e. higher extreme values). Differences in the location parameters can be associated to the bias between extreme values in the datasets. The scale parameters are normalized over the corresponding location parameters to appreciate the relative differences. Normalized scale parameters from HRC and CHRC are similar and lower than the ones derived from radar. Normalized scale parameters, together with their variability, tend to increase when moving from sea to Mediterranean, semiarid and arid climate, the larger the dispersion of the GEV distribution. A slight increase of the normalized scale parameters with duration can be noticed in the HRC/CHRC data".

6b) The figures axis could be improved if the authors want to make it clear to the reader. Also, I suggest the same size intervals in Y-axis (e.g., Fig 2&4).

[We moved here one of the specific comments of the reviewer that concerned the same issue: It is also suggested to present the parameter variation against the time by a linear representation that could help the readers to grasp the message from the figures.]

Thanks for the suggestion. The figure was revised. A linear scale on the duration axis is not recommended due to the almost logarithmically equi-spaced differences between the examined durations. Instead we used a logarithmic scale in the duration axis in order to improve the information provided by the figure and included the x-axis labels in all panels. Caption was then updated accordingly: "*The parameters for different products are represented around the corresponding duration so the logarithmic scale in x-axes should be interpreted accordingly*"

7) [Moved above]. We kindly refer to the reply to comment 3.

8) Some unnecessary materials: in my opinion, it is completely unnecessary to have Figure 9, when it is already well-known that the uncertainties increase for longer return period.

[We move here one of the specific comments of the reviewer that concerned the same issue: - Why Fig 9 presented? It is well known that confidence in a return level decreases rapidly when the period is more than about two times the length of the original data set. Fig 9 could be removed.]

Uncertainty increases with return period. As underlined by the reviewer, this is well known and expected. However, our objective is to quantify the uncertainty due to the use of a reduced record for different climates and different record lengths. In fact, the information provided by Fig. 9 and 10 is more than "uncertainties increase for longer return period": it includes (i) quantitative information (ii) for different climates and (iii) for different record lengths. Fig. 9 only reports 3 examples among many, but we think is it useful for the reader to better understand the message underlying Fig. 10.

9) Finally, I would like to draw your attention to a somewhat old, but still relevant, review on methods to IDF and DDF curves and their uncertainties that might have some valuable lessons in it: Aart Overeem, Adri Buishand and Iwan Holleman, Journal of Hydrology, 2008, 348, pp 124-134 We thank the reviewer you for the suggestion. The suggested paper, actually still up-to-date, helped in this revision process and is now cited in the new version of the manuscript.

**Specific Comments:**

Page 5- Line 15:

- I would expect to introduce parameters and explain a little bit how they were computed.

- It is not clear which method used for estimating the GEV distribution parameters by maximum likelihood.

- The assumptions regarding the convenience of the GEV to represent the AMS distribution through the Fisher-Tippett theorem, as well as calculation procedures, would be useful to present in the Appendix.

We thank the reviewer for these comments. We would like to point out that we used an established extreme values distribution (GEV) and derived the distribution parameters using one of the most commonly used methods (maximum likelihood). Details on these methods can be found in textbooks (e.g. Coles, 2001) and in the references provided in the text. Our contribution does not advance knowledge on this aspect.

As the reviewer suggested, the parameters need to be introduced in the main text. Therefore, we updated text in section 3.1 as follows: *"The GEV distribution is a three parameters extreme values distribution used worldwide to model rainfall extremes (e.g. Fowler and Kilsby, 2003; Gellens, 2002; Koutsoyiannis, 2004; Overeem et al., 2008). It is described by the location, scale and shape parameters, representing*

mean, dispersion and skewness of the distribution, respectively. [...] At-site GEV parameters (i.e. pixel by pixel) were identified using the maximum likelihood method (MATLAB statistics toolbox)."

As suggested, Appendix was revised to include information on the convergence of extreme values distributions to the GEV: "Under hypotheses on the regularity of the tail of the distribution, the Fisher-Tippet theorem demonstrates that GEV distribution is the only possible limit distribution for the extreme values of independent and identically distributed random variables."

- The bootstrap was applied to assess this uncertainty of record length. This method considers only the uncertainty for the return periods (it's my understanding). What about the uncertainty of the estimation of the GEV parameters (i.e. sampling errors)?

The MATLAB statistics toolbox used in this study provides the confidence interval of the derived GEV parameters. However, the uncertainty of estimation of the GEV parameters is not the subject of this study. Overeem et al. (2008) and Overeem et al. (2009) have proposed the bootstrapping of the data series used for the GEV fit to identify the uncertainty related to the fit. Perhaps this approach used in the past motivates the question by the referee.

In sections 3.3 and 4.4 of this study, we are assessing the uncertainty related to the record length, i.e. to the under-sampling of rainfall climatology. To do so, as mentioned by the reviewer, we devised the bootstrap approach based on long records of rain gauge data, that are here assumed to represent the variability of extreme rainfall climatology (section 3.3: "*We assumed the records of rain gauge data to be a complete sample of the climatology of extremes for return periods comparable to the remotely sensed data record length*"). The bootstrapped approach consists in three steps: (i) randomly sampling the AMS, (ii) deriving the GEV parameters and (iii) deriving the IDF values for selected return periods. The record length-related uncertainty in the GEV parameters is implicitly calculated (as 5-95th quantile interval) as a necessary step of the bootstrapping.

**Page6- Line 12:**

- Figure 2: Estimates of the GEV parameters could be present by Box-Whisker plot that obtained from 999 bootstrap samples, as well for 25th and 75th percentiles of the bootstrap samples.

We think the reviewer misunderstood Fig. 2. The figure shows the 25-75th quantile interval of the set of parameters obtained from pixels characterized by the examined climate, i.e. the 25-75th quantile interval among the number of pixels reported in Table 1. No bootstrap is performed here. We prefer to keep the quantile interval bars rather than using a box plot (i) because the number of elements in each climatic class is different and (ii) for consistency with Fig. 3 and with the use of quantile intervals we adopted throughout the paper.

It is also suggested to present the parameter variation against the time by a linear representation that could help the readers to grasp the message from the figures.

[Moved above]. We kindly refer to comment 6b above for more information.

- How are different climatic regions thought to affect precipitation (location and scale parameters)? It should be discussed at least (regarding figure 2).

Thank you for making this point. We revised section 2.1 by inserting the following statement: "Important gradients have been reported also for the climatology of extreme rainfall. Low return period intensities were found to be scaled with the mean annual precipitation. Conversely, the more arid the climate is, the more skewed the extreme value distribution is, with long return period intensities for arid areas being higher than the corresponding values for semiarid and Mediterranean areas, especially for short durations (Ben Zvi, 2009; Marra and Morin, 2015)."

**Page 6-**

- The location and scale parameters are not addressed in details. It is not clear to the reader without background knowledge of the GEV df.

Thank you for making this point. We added text to recall the meaning of the parameters and to be more explicit: "We recall here that the scale, location and shape parameters provide a measure of the mean, dispersion and skewness of the underlying distribution, respectively. Location parameters from HRC (CHRC) estimates are smaller (larger) than the ones from radar over Mediterranean climate and over the sea, meaning that extreme values from HRC (CHRC) are in general lower (higher) than radar extreme values while in semiarid and arid climates HRC and CHRC generally identify higher parameters than the radar (i.e. higher extreme values). Differences in the location parameters can be associated to the bias between extreme values in the datasets. The scale parameters are normalized over the corresponding location parameters in order to appreciate the relative differences. Normalized scale parameters, together with their variability, tend to increase when moving from sea to Mediterranean, semiarid and arid climates. The drier climate, the larger the dispersion of the GEV distribution. A slight increase of the normalized scale parameters with duration can be noticed in the HRC/CHRC data."

**Page 8-**

**- I am totally lost. What's the link between section 4.4 with previous sections?**

This section presents results on the quantification of uncertainty related to the record length of remote sensing datasets, introduced in section 3.3. To make things clearer we added a short introductory sentence to the section: "In this section, we present the results of the bootstrap sampling of long rain gauge records used to quantify the uncertainty related to the record length of remote sensing datasets. The uncertainty presented here is the component related to the under-sampling of rainfall climatology due to the use of short data records and is quantified as the 5–95th quantile interval of the bootstrap sampling."

**Page 12- Line 13:**

- Sorry if I missed this, how is the coverage of upscaled radar and CMORPH pixels? Are they exactly the same? Please, explicitly specify their spatial coverage.

Page 12 actually contains references, so we are not sure we understood what the reviewer is referring to (page 7?). We think that the reviewer can be either asking about each "upscaled" radar and CMORPH pixel, information that can be found in section 3.1 (page 5, lines 11-12): "In order to have radar and satellite data on a common grid suitable for the comparison, the full archive of hourly radar data was remapped by spatially averaging the  $1 \times 1$  km2 radar pixels to the corresponding ~8×8 km2 CMORPH pixels" or about the areal coverage of the comparison, which is reported in section 3.2 (page 5, lines 24-27): "The comparison is extended over an analysis domain defined excluding the pixels that are known to be not reliable. In particular, pixels located closer than 27 km or farther than 185 km from the radar or behind the hilly region are excluded due to insufficient reliability of the radar data, and pixels located in proximity of major lakes are excluded due to false rainfall signals in the CMORPH rainfall estimates".

**Page 18-**

- What's the dashed line in Fig 2 for shape parameter? The dashed lines showed the "0" value. We propose to remove it in order to make the figure clearer.

Page 20-

- Appendix A. The legend and description in caption do not show similar things (check).

We guess this comment also refers to both Fig. 4 and 6. We apologize for this; the captions will be updated.

**Page 25-**

- Why Fig 9 presented? It is well known that confidence in a return level decreases rapidly when the period is more than about two times the length of the original data set. Fig 9 could be removed.

[Moved above]. Please, see our response to comment 8 above.

**References**

- Borga, M., E. N. Anagnostou, G. Blöschl, and J. D. Creutin. 2011. Flash Flood Forecasting, Warning and Risk Management: The HYDRATE Project. Environmental Science and Policy 14(7): 834–44
- Borga M., M. Stoffel, L. Marchi, F. Marra, M. Jacob 2014: Hydrogeomorphic response to extreme rainfall in headwater systems: flash floods and debris flows. Journal of Hydrology, 518 (2014), 194–205. http://dx.doi.org/10.1016/j.jhydrol.2014.05.022
- Coles, S., 2001. An Introduction to Statistical Modeling of Extreme Values. Springer-Verlag, London.
- Overeem, A., Buishand, A., Holleman, I., 2008. Rainfall Depth-Duration-Frequency Curves and Their Uncertainties. J. Hydrol. 348(1–2): 124–34. http://dx.doi.org/10.1016/j.jhydrol.2007.09.044
- Panziera L., M. Gabella, S. Zanini, A. Hering, U. Germann and A. Berne, 2016. A radar-based regional extreme rainfall analysis to derive the thresholds for a novel automatic alert system in Switzerland. Hydrol. Earth Syst. Sci., 20, 2317–2332. http://dx.doi.org/10.5194/hess-20-2317-2016
- Segoni, S., Battistini, A., Rossi, G., Rosi, A., Lagomarsino, D., Catani, F., Moretti, S., and Casagli, N.: Technical Note: An operational landslide early warning system at regional scale based on space– time-variable rainfall thresholds, Nat. Hazards Earth Syst. Sci., 15, 853-861, http://dx.doi.org/10.5194/nhess-15-853-2015
- Svensson, C., and D. A. Jones. 2010. Review of Methods for Deriving Areal Reduction Factors. J. Flood Risk Management 3 3: 232–45. http://dx.doi.org/10.1111/j.1753-318X.2010.01075.x
- Tiranti, Davide et al. 2014. "The DEFENSE (Debris Flows triggEred by Storms Nowcasting System): An Early Warning System for Torrential Processes by Radar Storm Tracking Using a Geographic Information System (GIS)." Computers and Geosciences 70: 96–109. http://dx.doi.org/10.1016/j.cageo.2014.05.004
- Villarini, Gabriele et al. 2010. "Towards Probabilistic Forecasting of Flash Floods: The Combined Effects of Uncertainty in Radar-Rainfall and Flash Flood Guidance." Journal of Hydrology 394(1– 2): 275–84. http://dx.doi.org/10.1016/j.jhydrol.2010.02.014

---

## Author Comment (AC2) · 12 Jan 2017

**Manuscript reference number: hess-2016-597 - Response to RC2**

We would like to thank the reviewer for his/her to-the-point comments. We provide here our response and modifications introduced to the manuscript to address the reviewer's comments and improve the general presentation of the study. Below, in black text font is the reviewer's comments, and in blue is our response.

The manuscript "Comparing Intensity–Duration–Frequency curves derived from CMORPH and radar rainfall estimates over the Eastern Mediterranean" represents a good contribution to the assessment of satellite-based rainfall estimations in areas where their impact could be potentially relevant. The authors have done a thorough analysis over a sufficient amount of time and thus the results are significant.
The manuscript certainly fits the journal scope and is suggested for publication after the following points are considered by the authors:
We would like to thank the reviewer for his/her positive review.

1) The reason why both versions (gauged and ungauged) of CMORPH are used is not sufficiently explained and detailed in the text.
Thank you for raising this important issue – also shared by reviewer 1. Our introduction failed to emphasize the importance of using both the gauge-adjusted and un-adjusted versions of the satellite product (CMORPH, is this case). The gauge adjustment is expected to improve satellite estimates on average at the cost of having this improved product with several days of latency, which does not allow the use in real time applications. Furthermore, we would like to mention that there are regions on earth where there are no ground based sensors to adjust the satellite precipitation datasets. Hence, the natural question by the reviewers: 'why are you using the non-adjusted product in an application that is based on historical data rather than real time estimates?' The answer lies in the different requirements of the applications that make use of IDF curves: hydrologic design needs optimal quantitative accuracy, so evaluating unadjusted data is important for demonstrating uses of satellite precipitation in ungauged areas; early warning systems need to correctly identify the frequency of near real-time estimates. We updated text in the introduction to better motivate our study: *"[…] early warning systems, e.g. for flash floods (Borga et al., 2011; Villarini et al., 2010; Borga et al., 2014), landslides/debris flows (Tiranti et al. 2014; Borga et al., 2014; Segoni et al., 2015) or heavy rain (Panziera et al. 2016), need to operate in real time and rely on short-latency remote sensed measurements. In these situations, calculating the frequency of near real-time estimates using IDF curves derived from gauge-adjusted data could provide misleading results. It is therefore useful to analyse the characteristics of IDF curves derived from non-adjusted rainfall data, which are expected to represent the frequencies of near real time estimates."*, and we recalled the concept in the conclusions, adding a new point: *"Comparison of HRC-IDF and CHRC-IDF against radar-IDF show consistent patterns of correlation and dispersion, and different biases. This means that gauge-adjustment influences the magnitude rather than the space-time organization of annual extremes and suggests that HRC-IDF can potentially be used to estimate the frequencies of CMORPH estimates in near real time early warning systems."*
In addition to this, it is interesting to note that the performance of gauge-adjustment on extremes is affected by the different measurement scales of remote sensing instruments and rain gauges. Moreover, the impact of gauge-adjustment over ungauged areas is rarely quantified, especially for extremes and potentially introduces non homogeneity in space due to different gauge density data.

2) Did the authors check how many of the gauges are used in the production of the gauge-adjusted version of CMORPH in the area? This could have an impact in the analysis, maybe small. However, a couple of words on the subject need to be included.

Thank you for the question. The gauge-adjusted CMORPH product is using limited gauge data from the region, which may be also used in the adjustment of the radar-rainfall dataset. From Fig. 1d in Chen et al. (2008), the number of CPC gauges in the study area should be around ~12. We have contacted the developers of the gauge-adjusted CMORPH product for more specific information, but we have not received a response on our inquiry yet. We will mention this aspect in section 2.3: "*The gauge-adjusted CMORPH product is using data from ~12 gauges in the region (Chen et al., 2008), which may also be used in the adjustment of the radar-rainfall dataset.*"

3) While it is true that gauges are not available over the sea (!) the authors should also spend a few words on the fact that normally satellite-based estimations are far better over the sea surface. This partially contradicts their results. Just pay attention to this important fact. See, for example: Kidd, C., and V. Levizzani, 2011: Status of satellite precipitation retrievals. Hydrol. Earth Syst. Sci., 15, 1109-1116 for a discussion on the various methods of rain estimation from satellite.

We would like to thank the reviewer for this suggestion. We think that our results and text do not contradict his/her point. The results show that the CC between satellite and radar IDF maps over the sea is low. The text associates this to the absence of rain gauges over the sea. Since (i) gauge-adjustment is a crucial part of the radar quantitative precipitation estimation and (ii) similar CC patterns for HRC and CHRC are observed over land, we think that satellite datasets are expected to provide more accurate information on the spatial distribution of IDFs over the sea. We believe there has been a slight misunderstanding on the interpretation of this sentence, so we revised text to improve its clarity, also included a reference to the suggested paper: "*As pointed out above, gauge adjustment is only weakly impacting the space-time organization of CMORPH extreme estimates, while it is a crucial step in radar quantitative precipitation estimation. This observation, together with the increased reliability of satellite based estimations over the sea (Kidd and Levizzani, 2011), suggests that spatial distribution of IDF values indicated by satellite products should be considered more accurate.*"

4) A fine combing of the English is suggested since many imperfections are detected throughout the text.
English language has been re-checked and improved, thank you.

Minor point: The caption of Fig. 4 does not match the real colors used in the figure. It appears that the authors have used two different version prior and after a change they did while writing.
The reviewer guessed correctly. We apologize for the inconvenient, that affected also Fig. 6. The captions have now been updated.

*References*

Borga, M., E. N. Anagnostou, G. Blöschl, and J. D. Creutin. 2011. Flash Flood Forecasting, Warning and Risk Management: The HYDRATE Project. Environmental Science and Policy 14(7): 834–44

Borga M., M. Stoffel, L. Marchi, F. Marra, M. Jacob 2014: Hydrogeomorphic response to extreme rainfall in headwater systems: flash floods and debris flows. Journal of Hydrology, 518 (2014), 194–205. http://dx.doi.org/10.1016/j.jhydrol.2014.05.022

Chen, M., W. Shi, P. Xie, V. B. S. Silva, V. E. Kousky, R. Wayne Higgins, and J. E. Janowiak (2008), Assessing objective techniques for gauge-based analyses of global daily precipitation, J. Geophys. Res., 113, D04110, doi:10.1029/2007JD009132

Panziera L., M. Gabella, S. Zanini, A. Hering, U. Germann and A. Berne, 2016. A radar-based regional extreme rainfall analysis to derive the thresholds for a novel automatic alert system in Switzerland. Hydrol. Earth Syst. Sci., 20, 2317–2332. http://dx.doi.org/10.5194/hess-20-2317-2016

Segoni, S., Battistini, A., Rossi, G., Rosi, A., Lagomarsino, D., Catani, F., Moretti, S., and Casagli, N.: Technical Note: An operational landslide early warning system at regional scale based on space–

time-variable rainfall thresholds, Nat. Hazards Earth Syst. Sci., 15, 853-861, http://dx.doi.org/10.5194/nhess-15-853-2015

Tiranti, Davide et al. 2014. "The DEFENSE (Debris Flows triggEred by Storms - Nowcasting System): An Early Warning System for Torrential Processes by Radar Storm Tracking Using a Geographic Information System (GIS)." Computers and Geosciences 70: 96–109. http://dx.doi.org/10.1016/j.cageo.2014.05.004

Villarini, Gabriele et al. 2010. "Towards Probabilistic Forecasting of Flash Floods: The Combined Effects of Uncertainty in Radar-Rainfall and Flash Flood Guidance." Journal of Hydrology 394(1–2): 275–84. http://dx.doi.org/10.1016/j.jhydrol.2010.02.014

---

## Referee Report (RR1)

**Comparing Intensity-Duration-Frequency curves derived from CMORPH and radar rainfall estimates over the Eastern Mediterranean**

F. Marra et al.

I think that this is a very relevant paper because it compares the statistics of extremes for independent instruments over a climatic region characterized by strong gradients in precipitation patterns, for several temporal and spatial aggregations. The authors present a complete set of analyses showing the value of the extreme value theory applied to remotely sensed precipitation measurements available for a relative short period.

I have only minor comments and therefore I strongly encourage publication in HESS.

**Minor comments**

1) The rain gauges used to adjust satellite measurements are the same used in this study?

2) pag. 5 line 10. Can you provide an estimate of the amount of missing data? Do you know if some of your annual maxima from one instrument were measured when the others instruments were not working? This could potentially lead to a bias in the results, and, if it happened, should be mentioned in the text. Moreover, the sentence "In order to focus on a unique set of IDFs for each dataset, .." it is rather unclear to me.

3) Pag. 5 line 12: Since most of the rainfall occurs in winter and you are using calendar years, did you set a minimum time lag between close annual maxima occurred in separate years (e.g. if max rainfall occurs on 31 Dec and 1 Jan of following year, do you take them as two separate maximum)? Since your events should be independent according to the GEV theory, could you comment on this?

4) Pag. 24, line 23. The difference behavior of shape with temporal aggregation among satellite and radar should probably investigated a bit more. You state that the larger values of shape for the radar might be due to radar shutdowns, so you should probably check how many annual maxima from satellite were not measured by the radar because of the radar shutdowns.

5) Pag. 7 line 6: The sentences *"This means that the smoothing effect due to spatial and temporal aggregation of rainfall measurement depends on the return period, and is more pronounced for longer return periods. This relates to a non-homogeneity of the scales of rainfall extremes with return period: the more extreme an event is (.i.e. the longer its return period), the more localized it is expected to be in both space and time"* need some additional clarifications.

For satellites, your shape parameters were not decreasing with temporal aggregation, as shown in figure 2. Thus, the decrease in shape with temporal aggregation that you observe for radar (in figures 2 and 3) could not only be due to the smoothing effect, otherwise you should notice it also for the satellite in figure 2. It is true that the more extreme an event is, the more localized it is expected to be in both space and time, but in this study an event is defined according to a given spatio-temporal scale and not in absolute terms. Thus, I don't think it is correct to state that the longer is the return period, the more localized is the event in both space and time, because the return period is also relative to a given spatio-temporal scale. If your inhomogeneity of scales was true, than it would mean that we should have smaller shapes for longer spatial and temporal aggregations, but, for example, you did not get this for temporal aggregations for satellite. I am wondering if it is not just a

matter of archive length, i.e. within short archives it is less probable to observe the very heavy cases for large spatial and long temporal scales than for smaller scales. I think these aspects should be clarified a bit more in the text.

6) Shouldn't figure 4 be called return level plot and not IDF according to the standard nomenclature?

7) Pag. 7, line 23: do you have an idea why the spatial variability of the return levels values increases with return period?

8) Pag. 8 line 8: *"this reflects what observed……i.e. to predict larger intensities for longer return periods"*. I think you should add "with respect to longer durations", otherwise the sentence is rather obvious.

9) Table 1. I suggest to write: Number of pixels analyzed for each climatic region according to Koppen-Geiger classification. Are $T_{hot}$ and $T_{cold}$ the average monthly temperatures? Please specify.

---

## Author Response (AR2)

**Manuscript reference number: hess-2016-597 - Response to reviewers**

**Anonymous referees #1 and #2**

5  We would like to thank the referees for having reviewed the paper again. We are glad they are satisfied with the updates.

**Anonymous referee #3**

10  We thank the referee for the review, very helpful for a further improvement of the manuscript. Below our response (blue font) to the reviewer's comments (black).

I think that this is a very relevant paper because it compares the statistics of extremes for independent instruments over a climatic region characterized by strong gradients in precipitation patterns, for several
15  temporal and spatial aggregations. The authors present a complete set of analyses showing the value of the extreme value theory applied to remotely sensed precipitation measurements available for a relative short period.
I have only minor comments and therefore I strongly encourage publication in HESS.
Thank you for the positive words. We believe the referee had access to the discussion paper rather than
20  its updated form. However, there is no conflict with the comments addressed during the previous review.

1) The rain gauges used to adjust satellite measurements are the same used in this study?
We made use of data from 11 rain gauges. As reported in the updated manuscript (Section 2.3), "*the gauge-adjusted CMORPH product is using gauge data from ~12 gauges in the region (Chen et al.,*
25  *2008)*". There is a chance that some of the gauges used for the study are also included in the CMORPH adjustment procedure, but the authors couldn't gather further information on this.

2) pag. 5 line 10. Can you provide an estimate of the amount of missing data? Do you know if some of your annual maxima from one instrument were measured when the others instruments were not working?
30  This could potentially lead to a bias in the results, and, if it happened, should be mentioned in the text. Moreover, the sentence "*In order to focus on a unique set of IDFs for each dataset, ..*" it is rather unclear to me.
The area presents dry summers, during which the radar is turned off. It is usually turned back on as the first rain of the new season in expected (or, sometimes, observed). As reported by Marra and Morin
35  (2015), the long-term rain gauge data for the area are also characterized by lack of information (e.g. in some cases the zero rain information is not reported, so that it is equivalent to no-data, in other cases the gauges are known to have missed data but no information on when the data is missing is available). Precise quantification of the missed events is not feasible. We recognize this could potentially bias the results, and it is among the rationale for not imposing co-occurrence of the annual maxima and not cross-
40  checking the dates of occurrence of the annual maxima. We updated the text to make this aspect clearer: "*All the available rainfall estimates have been included in the IDF estimation, even if data from the other sources was missing during a given storm. No co-occurrence of the annual maxima is thus imposed. There*

*are, in fact, potential situations in which radar or rain gauges missed storms due to technical problems or power issues that cannot be directly identified from the data itself (Morin et al., 1998; Ben Zvi, 2009; Marra and Morin, 2015).*"

3) Pag. 5 line 12: Since most of the rainfall occurs in winter and you are using calendar years, did you set a minimum time lag between close annual maxima occurred in separate years (e.g. if max rainfall occurs on 31 Dec and 1 Jan of following year, do you take them as two separate maximum)? Since your events should be independent according to the GEV theory, could you comment on this?

Good point! We updated the text providing this information: "*A minimum time lag of 48 h between annual maxima observed in different years was set to fulfil the independency requirements of the GEV theory.*"

4) Pag. 24, line 23. The difference behavior of shape with temporal aggregation among satellite and radar should probably investigated a bit more. You state that the larger values of shape for the radar might be due to radar shutdowns, so you should probably check how many annual maxima from satellite were not measured by the radar because of the radar shutdowns.

Thank you for the comment. The difference observed in the shape parameters derived from radar/satellite estimates, as well as from radar/rain gauges reported by Marra and Morin (2015), could be ascribed to a number of potentially contributing causes. Separating between them is a difficult task that would require a dedicated study. Here, we mentioned the radar record intermittency, among others, because of its direct relation with the used data, and we mentioned other possible causes related to the CMORPH estimates ("*It should also be noted that missing of short duration extremes by CMORPH due to the overpasses frequency of microwave satellites could contribute to the differences observed between CMORPH and radar*"). To improve the presentation on this aspect, we modified the sentence to: "*Among the many possible causes, this can be associated…*".

However, we think that what the referee suggests (i.e. to check how many annual maxima from satellite were not measured by the radar) can be misleading: even if CMORPH data record was complete, it should not be used as a reference since there is no guarantee of a matching between MW satellite overpasses and rainfall events, particularly in arid areas. This would open a host of uncertainty sources.

5) Pag. 7 line 6: The sentences "*This means that the smoothing effect due to spatial and temporal aggregation of rainfall measurement depends on the return period, and is more pronounced for longer return periods. This relates to a non-homogeneity of the scales of rainfall extremes with return period: the more extreme an event is (.i.e. the longer its return period), the more localized it is expected to be in both space and time*" need some additional clarifications.

For satellites, your shape parameters were not decreasing with temporal aggregation, as shown in figure 2. Thus, the decrease in shape with temporal aggregation that you observe for radar (in figures 2 and 3) could not only be due to the smoothing effect, otherwise you should notice it also for the satellite in figure 2. It is true that the more extreme an event is, the more localized it is expected to be in both space and time, but in this study an event is defined according to a given spatio-temporal scale and not in absolute terms. Thus, I don't think it is correct to state that the longer is the return period, the more localized is the event in both space and time, because the return period is also relative to a given spatio-temporal scale. If your inhomogeneity of scales was true, than it would mean that we should have smaller shapes for

longer spatial and temporal aggregations, but, for example, you did not get this for temporal aggregations for satellite. I am wondering if it is not just a matter of archive length, i.e. within short archives it is less probable to observe the very heavy cases for large spatial and long temporal scales than for smaller scales. I think these aspects should be clarified a bit more in the text.

This is a good point. Summarizing: (a) the CMORPH shapes in Fig. 2 don't change with duration; (b) in the study an event is defined according to a given scale and not in absolute terms and the return period is also relative to that given scale; what observed could be a matter of record length.

(a) This is true. Our reasoning for using the radar for this analysis is driven by the idea that, owing to its quantitative retrieval method (low frequency of the MW overpasses + advection and morphing from VIS/IR information), CMORPH is arguably less accurate in providing space-time scale information. Text in section 3.1 is updated to provide this information: "*In this analysis, radar is preferred over CMORPH due to its ability to provide more direct rainfall estimates at the small spatial and temporal scales.*"

(b) We think that the word "event" was misused as it sounded referring more to single events rather than what was actually intended, the general case. The pattern observed in Fig. 3 relates to the spatial-temporal scales of extreme precipitation typically analyzed using the so called Areal Reduction Factors (ARF). Our results support previous results on ARFs derived from radar datasets (Bacchi and Ranzi, 1996; Durrans, 2002; Allen and De Gaetano, 2005; Lombardo 2006; Overeem et al., 2010). However, it should also be noted that other studies found no clear dependency of the ARFs on return period (Wright et al., 2014). The scientific debate is still open and, we agree with the referee, other factors could contribute to what reported in Fig. 3, including the record length. We updated the text in order to make it clearer and to provide some useful references: "*This means that the smoothing effect due to the spatial and temporal aggregation of rainfall measurement depends on the return period, and is more pronounced for longer return periods. These results relate to the spatial-temporal scales of extreme precipitation, usually analysed using the areal reduction factors, and suggest a non-homogeneity of the scales of rainfall extremes with return period. When using higher spatial and temporal resolutions, it is more probable to observe, in the relatively short archive of radar data, higher extreme events, since they are likely to be more localized in both space and time. This supports previous findings on areal reduction factors derived using radar data (Bacchi and Ranzi, 1996; Durrans, 2002; Allen and De Gaetano, 2005; Lombardo 2006; Overeem et al., 2010). However, it should be noted that other studies reported no clear dependency of the areal reduction factors on return period (Wright et al., 2014).*"

6) Shouldn't figure 4 be called return level plot and not IDF according to the standard nomenclature?
As suggested, Figure 4 is a composite of "return level plots" shown for different durations. In the text we used the term IDF curves and IDF values in order to maintain in the notation the full functional dependency. Therefore, as we did not introduce the use of "return levels" in the text, we think it would be preferable to be consistent in the figure captions.

7) Pag. 7, line 23: do you have an idea why the spatial variability of the return levels values increases with return period?

It is due to the larger uncertainty associated to the return levels estimated for longer return periods. We added a comment specifying this: "…*the spatial variability of IDF values increases with return period, owing to the larger uncertainty associated to longer return periods.*"

8) Pag. 8 line 8: "this reflects what observed……i.e. to predict larger intensities for longer return periods". I think you should add "with respect to longer durations", otherwise the sentence is rather obvious.
Thank you for pointing this out, the sentence has been updated.

9) Table 1. I suggest to write: Number of pixels analyzed for each climatic region according to Koppen-Geiger classification. Are Thot and Tcold the average monthly temperatures? Please specify.
Thank you for the suggestion, the table caption has been updated.
Concerning the Thot and Tcold information, we relied on the definitions by Peel et al. (2007). Generally, monthly averages are used, but this information is not explicitly provided by these authors.

[revised manuscript text omitted]